# On the sensitivity of aerosol-cloud interactions to changes in sea surface temperature in radiative-convective equilibrium

Suf Lorian[1,2] and Guy Dagan[1]

[1]Fredy and Nadine Herrmann Institute of Earth Sciences, The Hebrew University of Jerusalem, Jerusalem, Israel
[2]The Racah Institute of Physics, The Hebrew University of Jerusalem, Jerusalem, Israel

**Correspondence:** Guy Dagan (guy.dagan@mail.huji.ac.il)

**Abstract.** Clouds play a vital role in regulating Earth's energy balance, and are impacted by anthropogenic aerosol concentration ($N_a$) and sea surface temperature (SST) alterations. Traditionally, these factors, aerosols and SST, are investigated independently. This study employs cloud-resolving, radiative-convective-equilibrium (RCE) simulations to explore aerosol-cloud interactions (ACIs) under varying SSTs. ACIs are found to be SST-dependent even under RCE conditions. Notably, changes in cloud radiative effects for both the longwave and shortwave radiations lead to a decrease in top-of-atmosphere (TOA) energy gain with increasing $N_a$. The changes in TOA shortwave flux exhibit greater sensitivity to underlying SST conditions compared to longwave radiation. To comprehend these trends, we perform a linear decomposition, analyzing the responses of different cloud regimes and contributions from changes in cloud's opacity and occurrence. This breakdown reveals that ice and shallow clouds predominantly contribute to the radiative effect, mostly due to changes in cloud's opacity, due to the Twomey effect, which is proportional to the baseline cloud fraction. Moreover, with an increase in $N_a$, we observe an increase in latent heat release at the upper troposphere associated with heightened production of snow and graupel. We show that this trend, consistently across all SSTs, affects the anvil cloud cover by affecting the static–stability at the upper troposphere via a similar mechanism to the iris–stability effect, resulting in an increase in outgoing longwave radiation. In conclusion, under the ongoing climate change, studying the sensitivity of clouds to aerosols and SST should be conducted concomitantly as mutual effects are expected.

## 1 Introduction

The response of clouds to anthropogenic perturbations is highly uncertain, posing a significant challenge in predicting future climate. This uncertainty stems mainly from two aspects: 1) uncertainty regarding the change in top-of-atmosphere (TOA) radiative flux resulting from the cloud response to warming, referred to as cloud feedback (Ceppi et al., 2017), and 2) uncertainty regarding the response of clouds to anthropogenic aerosols (Bellouin et al., 2020). In the latter case, aerosols, which can serve as cloud condensation nuclei (CCN) and ice nuclei, could affect the microphysical properties and processes in clouds (Bellouin et al., 2020). Specifically, clouds forming under higher aerosol concentrations (polluted clouds) usually have initially smaller and more numerous droplets, with a narrower size distribution compared to clean clouds (Squires, 1958; Squires and Twomey, 1960). The initial droplet size distribution affects the cloud's albedo (Twomey, 1974, 1977; Bellouin et al., 2020) and

can affect key cloud processes such as condensation–evaporation, collision–coalescence and sedimentation (Albrecht, 1989; Seinfeld et al., 2016; Dagan et al., 2017; Heikenfeld et al., 2019; Christensen et al., 2022). These effects are known to be dependent on the environmental conditions (Gryspeerdt and Stier, 2012; Christensen et al., 2016; Dagan and Stier, 2020b), hence are expected to be state/time-dependent under ongoing climate change (Dagan et al., 2017; Igel and van den Heever, 2021; Dagan, 2022).

Ultimately, the microphysical effects mentioned above could modify the precipitation production (Albrecht, 1989). Specifically, the initiation of warm rain has been shown to be delayed and to start at higher elevations under more polluted conditions (Rosenfeld, 2000; Freud and Rosenfeld, 2012; Dagan et al., 2015; Heikenfeld et al., 2019). However, in deep convective clouds, the precipitation production could be compensated — or even over-compensated — for at higher levels of the clouds to which more water is advected under more polluted conditions (Rosenfeld et al., 2008; Koren et al., 2014; Altaratz et al., 2014). As the freezing level elevation increases with sea surface temperature (SST), at lower SSTs the warm layer (containing liquid only) of a deep convective cloud is narrower in comparison to higher SSTs. Thus, an aerosol perturbation is hypothesized to more likely suppress warm rain completely at lower SSTs than at higher SSTs, where the relatively deep warm layer of the clouds enables longer diffusional growth of the droplets to the critical size which initiates precipitation (Freud and Rosenfeld, 2012; Heikenfeld et al., 2019). Warm rain suppression and, as a consequence, enhanced freezing of this water in the cold (containing ice) sections of the cloud, will result in more latent heat release at the upper parts of the troposphere (Rosenfeld et al., 2008; Igel and van den Heever, 2021) and thus in changes in the atmospheric stability.

In addition to the effect on precipitation, it has been previously suggested that the aerosol's effect on deep convective clouds can increase the anvil cloud mass and extent by increasing the upward advection of water (Fan et al., 2010, 2013; Grabowski and Morrison, 2016; Chen et al., 2017). This trend could be explained by the convective invigoration hypothesis (Williams et al., 2002; Koren et al., 2005; Seifert and Beheng, 2006; Rosenfeld et al., 2008; Yuan et al., 2011; Koren et al., 2014). Under this hypothesis, which remains highly questionable (Varble et al., 2023; Romps et al., 2023), increasing aerosol concentrations have been suggested to drive stronger latent heat release and hence stronger vertical velocities. In addition, under high aerosol concentration conditions, the smaller hydrometeors are transported higher into the atmosphere for a given vertical velocity (Koren et al., 2015; Dagan et al., 2018, 2020), and their lifetime at the upper troposphere is longer, due to a weaker sedimentation rate (Fan et al., 2013; Grabowski and Morrison, 2016). However, it is important to note that these proposed aerosol effects are still highly uncertain (Stevens and Feingold, 2009; Varble, 2018; Romps et al., 2023; Varble et al., 2023).

Cloud feedback, or the response of the cloud radiative effect (CRE) to surface warming, was recently shown to depend on the assumed aerosol concentration (Dagan, 2022). In the tropics, the radiative effect of both shallow (Gettelman and Sherwood, 2016) and deep (Ceppi et al., 2017) clouds is expected to further warm the surface. Shallow tropical and sub-tropical clouds — which have a general radiative cooling effect — are expected to become less prevalent and less radiatively opaque, thus producing a positive (but still highly uncertain) feedback (Gettelman and Sherwood, 2016; Nuijens and Siebesma, 2019). At the same time, deep tropical clouds are also expected to react to surface warming in a way that modifies their CRE (Ceppi et al., 2017). Specifically, it has been suggested that the tropical anvil cloud temperature and coverage react to surface warming

(Hartmann and Larson, 2002; Zelinka and Hartmann, 2010; Bony et al., 2016; Ceppi et al., 2017). Tropical anvil clouds strongly modulate the longwave emissions of Earth as these clouds are much colder than the surface (by about 70-90K) and are often opaque in the longwave, thus emitting a significantly lower amount of energy to space than otherwise would be emitted without them. In addition, anvil clouds could also strongly modulate the shortwave radiation budget, depending on their optical thickness (Hartmann and Berry, 2017; Li et al., 2019; Sokol et al., 2024). Hence, any anthropogenic-driven changes to the anvil cloud properties, such as amount and temperature, could significantly affect Earth's energy budget (Zelinka and Hartmann, 2010; Ceppi et al., 2017).

A central feature of the anvil cloud response to SST changes is the fixed anvil temperature (FAT) hypothesis (Hartmann and Larson, 2002), which states that the temperature of anvil clouds is anticipated to remain roughly fixed with warming. According to the FAT hypothesis, anvil top heights are determined by clear-sky radiative cooling, which in turn is primarily determined by water vapor concentration. The water vapor concentration, following the Clausius-Clapeyron relation, sharply drops to negligible values near the temperatures of the upper-troposphere, making the radiative cooling inefficient above this level and still efficient below this level. In a clear-sky free troposphere, radiative cooling is balanced by adiabatic warming due to subsiding motions, thus the energy budget can be formulated as follows:

$$Q_r = -S\omega \tag{1}$$

where $Q_r$ is the radiative cooling rate, $\omega$ is the clear sky vertical pressure velocity, and $S$ is the static-stability defined as:

$$S = -\frac{T}{\theta}\frac{\partial \theta}{\partial P} \tag{2}$$

where $T$ is the air temperature, $\theta$ is the potential temperature and $P$ is the pressure.

The subsidence motion below the sharp drop in radiative cooling and the lack of subsidence above this level generates vertical divergence in the clear sky, which, due to conservation of mass, is balanced by horizontal divergence from the convective regions. This convective divergence controls anvil clouds (Hartmann and Larson, 2002; Zelinka and Hartmann, 2011, 2010; Bony et al., 2016) (Below, in Fig. 7, vertical profiles of $S$, $Q_r$, $\omega$ and its vertical divergence are presented).

While observations, global climate models and high-resolution, convective-permitting models predict an increase in altitude of anvil clouds while maintaining nearly fixed temperatures, they also anticipate a decrease in anvil cloud coverage with rising surface temperatures (Zelinka and Hartmann, 2011; Bony et al., 2016; Williams and Pierrehumbert, 2017; Wing et al., 2020; Saint-Lu et al., 2020; Beydoun et al., 2021). The mechanisms behind this decrease in anvil cloud coverage rely on the same physics as do the mechanisms of the FAT hypothesis. Namely, it has been suggested that as the climate warms, the clouds rise, but find themselves in a more stable atmosphere (while remaining at nearly the same temperature). This enhanced stability under warmer conditions reduces the convective outflow in the upper troposphere and hence decreases the anvil cloud fraction (Bony et al., 2016; Beydoun et al., 2021). Specifically, it was shown that the maximum of the radiative-driven mass divergence in convective regions ($D_r$), defined as:

$$D_r = \frac{\partial \omega}{\partial P} \tag{3}$$

can accurately predict the anvil cloud fraction, and decreases with the increase in stability occurring with an increase in SST
(Bony et al., 2016; Beydoun et al., 2021). In addition to the radiative-driven divergence, slow evaporation (Seeley et al., 2019)
and sedimentation (Beydoun et al., 2021) of the ice crystals at the upper troposphere contribute to anvil cloud formation.
However, changes in the lifetime of anvil clouds — determined by changes in sedimentation and evaporation — were shown
to play a secondary role in the response of anvil clouds to warming (Beydoun et al., 2021).

In this study, we focus on the synergistic SST and aerosol effects on tropical convective clouds, and specifically on the
CRE, under equilibrium conditions using idealized cloud-resolving, radiative-convective-equilibrium (RCE) simulations. This
is done following previous studies that uses RCE to examine different aspects of ACI (van den Heever et al., 2011; Storer and
van den Heever, 2013; Beydoun and Hoose, 2019; Dagan, 2022).

## 2    Methods

### 2.1    Model description

The model used in this study is the System for Atmospheric Modeling (Khairoutdinov and Randall, 2003, SAM) version 6.11.7.
The microphysics scheme used is the two-moment bulk microphysics of Morrison et al. (2005). The aerosols available for
activation are represented by a power law function of the super-saturation ($SS$): CCN = $N_a SS^k$, where $N_a$ is the concentration
of CCN available at 1% super-saturation and $k$ is a constant, here equal to 0.4, representing typical maritime conditions. CCN
activation at the cloud base is parameterized using the vertical velocity and CCN spectrum parameters (Twomey, 1959). In this
case, we use different $N_a$ concentrations for representing changes in aerosol concentration. Here, ice nucleation is not directly
coupled to $N_a$ (i.e., changes in $N_a$ do not change the concentration of ice nucleating particles- INP), but rather depends on
the temperature and the supersaturation with respect to ice (Rasmussen et al., 2002). We note that, in realistic conditions,
changes in $N_a$ might cause changes in INP, an effect that should be addressed in future research. In our simulations, freezing
occurs through homogeneous freezing and heterogeneous freezing by contact or immersion freezing (Morrison et al., 2005).
Ice nucleation directly from vapor is not considered here, but depositional growth of cloud ice is enabled. Direct interactions
between aerosols and radiation are also not considered here, however, aerosols could affect the radiation via modifying the
clouds' properties. In order to represent the Twomey effect (Twomey, 1977), the model is configured to pass cloud water and
ice-crystal effective radii from the microphysics scheme to the radiation scheme.

### 2.2    Experimental design

The simulations used here generally follow the Radiative-Convective-Equilibrium Model Intercomparison Project (Wing et al.,
2018, RCEMIP) small domain protocol but with changes in aerosol concentration. The simulations are run in a small domain,
of $96 \times 96$ km$^2$, in order to avoid the effects of convective self-aggregation (Muller and Held, 2012; Lutsko and Cronin, 2018).
The simulations are conducted with a horizontal grid spacing of 1 km, 68 vertical levels between 25 m and 31 km, and a
vertical grid spacing increasing from 50 m at the surface to around 1 km at the domain top. To get solar insolation close to the

tropical-mean value, the solar radiation is fixed at 551.58 W m$^{-2}$, with a zenith angle of 42.05°(Wing et al., 2018). A diurnal cycle is not considered here, and we note that it might affect the convective development to some extent even over the ocean (Nesbitt and Zipser, 2003; Gasparini et al., 2022). In order to initialize convection, a small thermal noise is added near the surface at the beginning of each simulation.

The concentration of $CO_2$ is fixed at the pre-industrial level (280 ppm), while there are 25 different $N_a$ and SSTs combina-
tions, 5 different values for each. $N_a$ ranges from 20 to 2000 cm$^{-3}$ (20, 100, 200, 1000, and 2000 cm$^{-3}$), following a recent observational data set (Choudhury and Tesche, 2023), which showed the feasibility of this $N_a$ range. The SST ranges from 290 to 310 K in 5 K intervals. Snapshots of the different simulations is presented in Fig. S1, SI. This wide range of aerosol and SST conditions are used to maximize the effects and for establishing a better physical understanding. A fixed ozone profile, repre-senting a typical tropical atmosphere, is used here (Wing et al., 2018). We note that using a fixed ozone profile under different
SSTs is not entirely realistic and may have some effect on the clouds development (Harrop and Hartmann, 2012; Seidel and Yang, 2022). For simplicity, the effect of other trace gases (such as $CH_4$ and $N_2O$) is neglected. The temporal resolution of the simulations is 10 seconds, and of the interactive radiative scheme is 5 minutes (using the CAM radiation scheme (Collins et al., 2006)). All fields have an output resolution of 1 hour; 3-D fields are saved as snapshots, while domain statistics are saved as hourly averages. Each simulation was run for 150 days (Wing et al., 2018), and the last 30 days of each simulation were used
for statistical analysis.

## 3    Results and discussion

### 3.1    Response of the domain mean properties to aerosol perturbation under different SSTs

We start by examining the effect of changes in $N_a$ on the TOA energy gain under different SSTs ($\Delta R$; Fig. 1a). This figure illustrates that for all SSTs, an increase in $N_a$ decreases $\Delta R$, an effect which becomes stronger with a decrease in the SST.
Both the longwave (LW) and shortwave (SW) components of $\Delta R$ are negatively affected by $N_a$ (each declining up to 4–5 W m$^{-2}$ for the entire $N_a$ range considered here, depending on the SST; Fig. 1b and c), with $\Delta R^{SW}$ being more susceptible to SST changes (Fig. 1c), and $\Delta R^{LW}$ decreases in a roughly similar manner across all SSTs (Fig. 1b). Moreover, the CRE (calculated as all sky radiative flux minus clear sky radiative flux) is identified as the main driver of $\Delta R$ variations, while changes in clear sky radiation has a minimal impact, as indicated by Fig. 1d-f. This is true in our simulations as changes in $N_a$ do not directly
affect radiation by aerosol-radiation interactions.

In order to understand the radiative effect of an increase in $N_a$ under the different SSTs, we first examine the domain- and time-mean cloud liquid water path, ice water path and cloud fraction ($\mathcal{L}$, $\mathcal{I}$ and CF respectively; Fig. 2). This Figure illustrates a monotonic increase in $\mathcal{L}$ with $N_a$, which is generally stronger under lower SSTs, and a monotonic decrease in $\mathcal{I}$, consistently across SSTs. In addition, Fig. 2c illustrates a general decrease in CF with $N_a$, although not monotonic. We note that the CF
trend is dependant on the choice of the cloud vs. clear-sky definition, as can be seen in Fig. S2, SI.

Next, we examine vertical profiles of the different hydrometeors (Fig. 3). We note that with an increase in SST, the freezing level increases. Since an increase in $N_a$ acts to push warm rain formation to higher levels (Rosenfeld, 2000; Freud and Rosen-

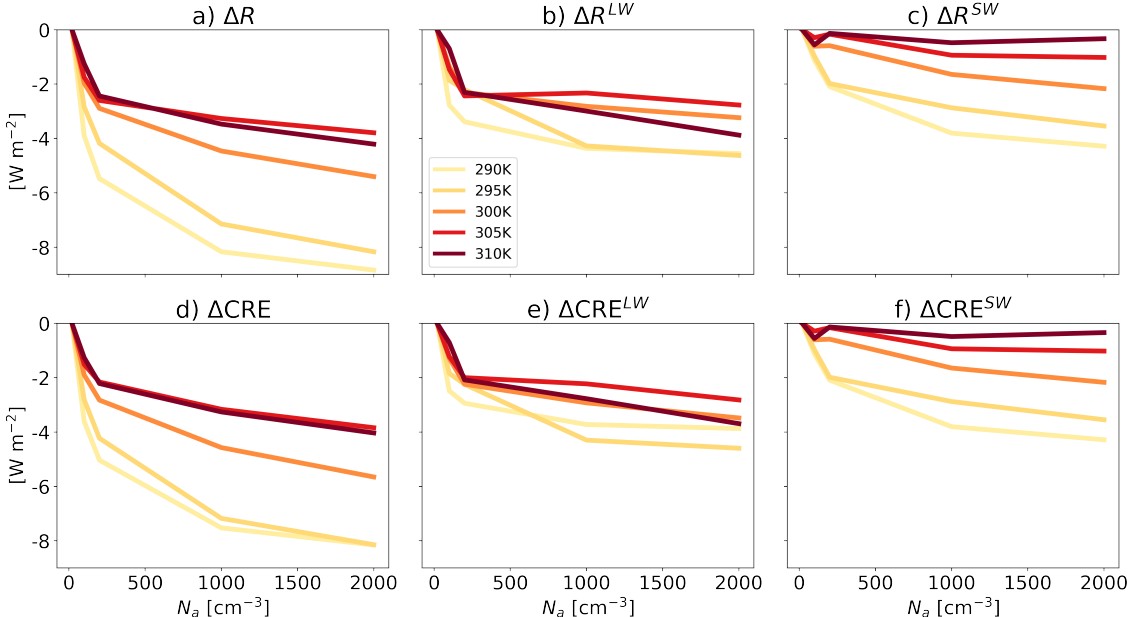

**Figure 1.** Changes in the domain and time mean radiative fluxes at the top of the atmosphere due to changes in aerosol concentrations ($N_a$). (**a**) presents the total change in radiation, while (**b**) and (**c**) present changes in longwave (LW) and shortwave (SW) radiation, respectively. (**d-f**) present the changes in the total cloud radiative effect (CRE) and its LW and SW components, respectively. The values are presented relative to the cleanest run ($N_a = 20$ cm$^{-3}$) for each SST, as indicated by the $\Delta$ sign.

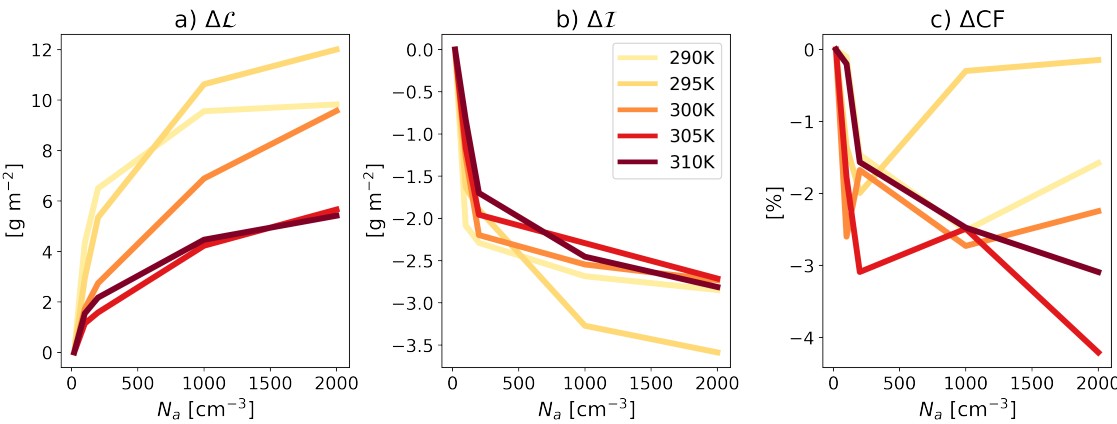

**Figure 2.** The response of domain and time mean liquid water path ($\mathcal{L}$; **a**), ice water path ($\mathcal{I}$; **b**) and cloud fraction (CF; **c**) to an increase in $N_a$. The values are presented relative to the cleanest run ($N_a = 20$ cm$^{-3}$) for each SST, as indicated by the $\Delta$ sign.

feld, 2012; Heikenfeld et al., 2019), under lower SSTs, for which the freezing level is relatively shallow (about 1250 m above cloud base in the coldest case considered here), an increase in $N_a$ can inhibit warm rain (Fig. 3g). In contrast, under higher

SSTs, for which the freezing level is relatively deep (about 6000 m above cloud base in the warmest case considered here), an increase in $N_a$ drives warm rain inhibition at the lower levels, which is compensated for at higher levels of the warm section (Fig. 3g). That is to say that under low SSTs, the delay in warm rain is not being offset at higher levels within the warm section, while under high SSTs we do see such an offset. This explains the stronger rise in water content within the warm section ($\mathcal{L}$) with an increase in $N_a$ (Albrecht, 1989) under low SST conditions compared to high SST conditions (Fig. 2a).

Beside resulting in an increase in $\mathcal{L}$, the warm rain inhibition under higher $N_a$ results in more super-cooled water (Carrió and Cotton, 2011; Chen et al., 2017, Fig. 3f), leading to higher production of snow (Chen et al., 2017, Fig. 3j), and drives higher riming rates, thus producing more graupel (Chen et al., 2017, Fig. 3i). We will get back to this observed trend for the explanation of the results presented in Fig. 10 below. In addition, cloud ice declines with $N_a$, consistently across SSTs (Fig. 3h). This trend is consistent with the decline in $\mathcal{I}$ and CF (2b and c, respectively) and will be discussed further below (Fig. 6d).

## 3.2 Response by cloud regimes

Figs. 1, 2 and 3 examine the bulk cloud and radiative properties in the domain. However, as previously demonstrated, the impact of aerosols on clouds is cloud regime dependent (Gryspeerdt and Stier, 2012; Christensen et al., 2016; Dagan and Stier, 2020b). Therefore, it is crucial to analyze the distribution of cloud regimes in our simulations and discern how each specific cloud regime responds to the increase in $N_a$. In this paper we define the cloud regimes based on different bins of $\mathcal{L}$ and $\mathcal{I}$. For that purpose, Fig. 4 presents 2D histograms of the cloud occurrence (CO) at the different bins of $\mathcal{L}$ and $\mathcal{I}$, and the average total, shortwave and longwave CRE at these different bins, all for the coldest case considered here (SST = 290 K) as an example. This figure illustrates that the CF in these RCE simulations is mostly dominated by anvil clouds (e.g. Wing et al. (2020)), i.e., clouds with negligible $\mathcal{L}$ and high (thick anvil clouds; denoted by marker 1 in Fig. 4a) or low (thin anvil clouds; denoted by marker 2 in Fig. 4a) $\mathcal{I}$. However, Fig. 4a also illustrates the existence of two other types of clouds in these RCE simulations – shallow clouds (high $\mathcal{L}$ and low $\mathcal{I}$; denoted by marker 3 in Fig. 4a) and deep convective clouds (high $\mathcal{L}$ and high $\mathcal{I}$; denoted by marker 4 in Fig. 4a). We note that the shallow and deep cloud regimes may also consist of other types of clouds, such as cumulus congestus in the deep regime and two-layer cloud conditions with cirrus clouds with relatively low $\mathcal{I}$ above shallow clouds. Furthermore, Fig. 4e and j present the radiative significance of each $\mathcal{L}$ and $\mathcal{I}$ bin (i.e., the CO times the CRE for each bin), which illustrates a strong heating by thin anvil clouds and cooling by other cloud regimes. Lastly, Fig. 4 k-n illustrate the difference between simulations with the highest (2000 cm$^{-3}$) and the lowest (20 cm$^{-3}$) $N_a$ conditions. Specifically, Fig. 4k illustrates that an increase in $N_a$ drives thinning of the anvil clouds, i.e., an increase in the frequency of thin anvil clouds and a decrease in the frequency of thick anvil clouds. Additionally, Fig. 4 l-n illustrate that with an increase in $N_a$ the CRE decreases for all $\mathcal{L}$ and $\mathcal{I}$ bins (and especially for medium-high $\mathcal{L}$ and low $\mathcal{I}$; Fig. 4l), driven mostly by changes in the SW (Fig. 4m), with only minor changes in the LW (Fig. 4n). This SW difference with $N_a$ can be explained by the Twomey effect (Twomey, 1974).

Following the method outlined in Sokol et al. (2024), we calculate the total regime's CF as the 2D integral over the regime's $\mathcal{L}$ and $\mathcal{I}$ bins as defined in Table S1, SI. Fig. 5a illustrates a monotonic decrease across SSTs in thick anvil cloud fraction ($CF_{thick}$) with increasing $N_a$, consistently with the domain mean CF reduction (Figs. 2c and S8a, SI). On the other hand,

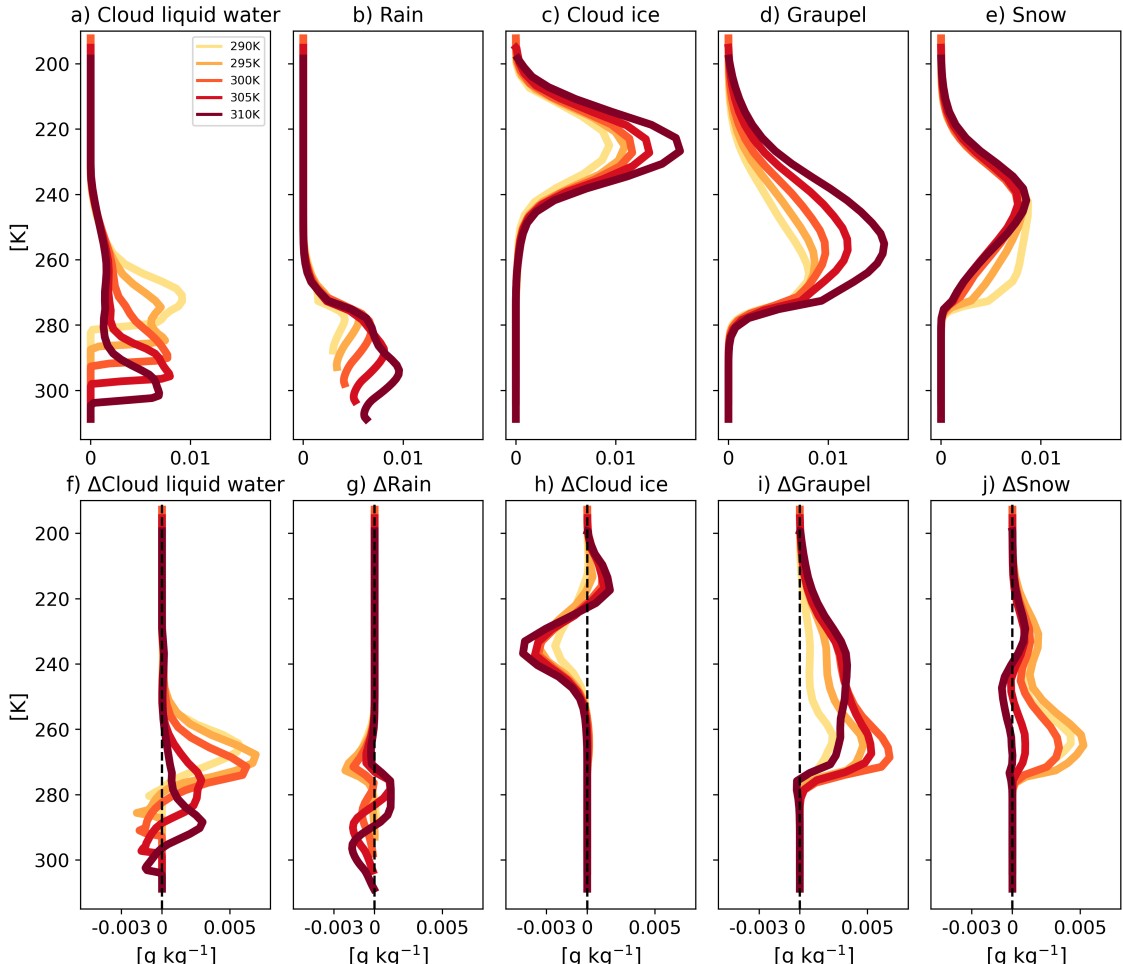

**Figure 3.** Domain and time mean vertical profiles of the different hydrometeors for the cleanest runs ($N_a = 20$ cm$^{-3}$): (**a**) cloud liquid water, (**b**) rain, (**c**) ice, (**d**) graupel, and (**e**) snow, and their response to increasing $N_a$ to 2000 cm$^{-3}$, relative to the cleanest run for each SST (**f** – **j**). Here we only present the cleanest and the response of the most polluted runs for clarity. The full range of $N_a$ is presented in Figs. S3-S7, SI.

thin anvil cloud fraction ($CF_{thin}$) mostly increases with $N_a$, generally stronger for lower SSTs (Fig 5b), which illustrates a general thinning of anvil clouds. We note that the entire distribution of $\mathcal{I}$ is shifted to lower values with $N_a$, demonstrating this thinning of the anvil clouds (Figs. 4k, 2b and 3h). A decrease in $CF_{thick}$ and thinning of the anvil clouds leads to more outgoing LW radiation out of the atmosphere and reduces $\Delta R^{LW}$, as can be seen in Fig. 1b. In addition, Fig. 5c presents the relative change in the shallow cloud fraction ($CF_{shallow}$). It illustrates a rise in $CF_{shallow}$ with $N_a$ for low SST, while for high SST it illustrates a decrease in $CF_{shallow}$ with $N_a$ (the change in the shallow cloud fraction is not observed in Fig. 4k due to the dominance of ice clouds, which inflates the color-bar range). We note that although the relative changes in $CF_{thick}$, $CF_{thin}$ and $CF_{shallow}$ have similar magnitudes, the baseline (i.e., referring to the simulated value, and not the difference between the

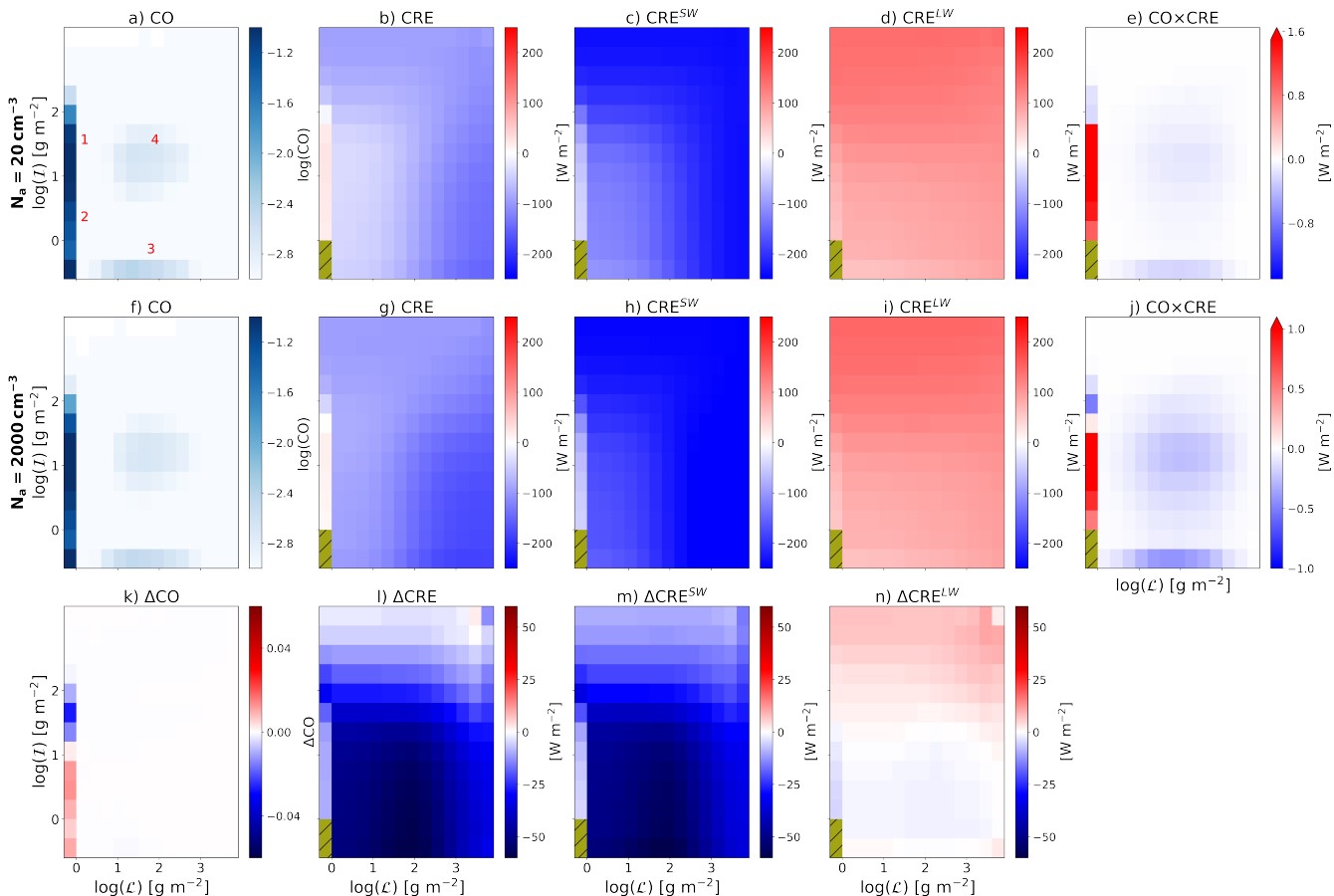

**Figure 4.** Domain and time mean two-dimensional histograms of cloud occurrence (CO; **a** and **f**), at different bins of liquid water path ($\mathcal{L}$) and ice water path ($\mathcal{I}$) and the average total (**b** and **g**), shortwave (**c** and **h**) and longwave (**d** and **i**) cloud radiative effect (CRE) at these different bins. Furthermore, the radiative significance (CRE×CO) of each bin is illustrated in panels **e** and **j**. These quantities are presented for two simulations using the lowest ($N_a = 20$ cm$^{-3}$; **a-e**), and the highest ($N_a = 2000$ cm$^{-3}$; **f-j**) $N_a$, under SST = 290 K. Four different cloud regimes are marked in red in panel **a**: (1) thick anvil clouds, (2) thin anvil clouds, (3) shallow clouds and (4) deep convective clouds, while the clear-sky regime is painted in tan. In addition, the difference between the highest and lowest $N_a$ conditions is presented in panels **k-n**.

most polluted and cleanest runs) $CF_{thick}$ and $CF_{thin}$ are an order of magnitude larger than $CF_{shallow}$ (Fig. S8, SI). Lastly, deep cloud fraction ($CF_{deep}$) changes in a non-monotonic trend with $N_a$ (Fig. 5d), while also covering a small fraction of the domain (Fig. S8d, SI).

Fig. 4 illustrates that the response of the CRE to an increase in $N_a$ is driven both by changes in CO (Fig. 4k) and by changes in CRE for a given bin of $\mathcal{L}$ and $\mathcal{I}$ (Fig. 4l). Next, we aim to quantitatively separate these two effects. Thus, we write the total CRE as the 2D integral over the different bins of $\mathcal{L}$ and $\mathcal{I}$ of the CF times the CRE in each bin:

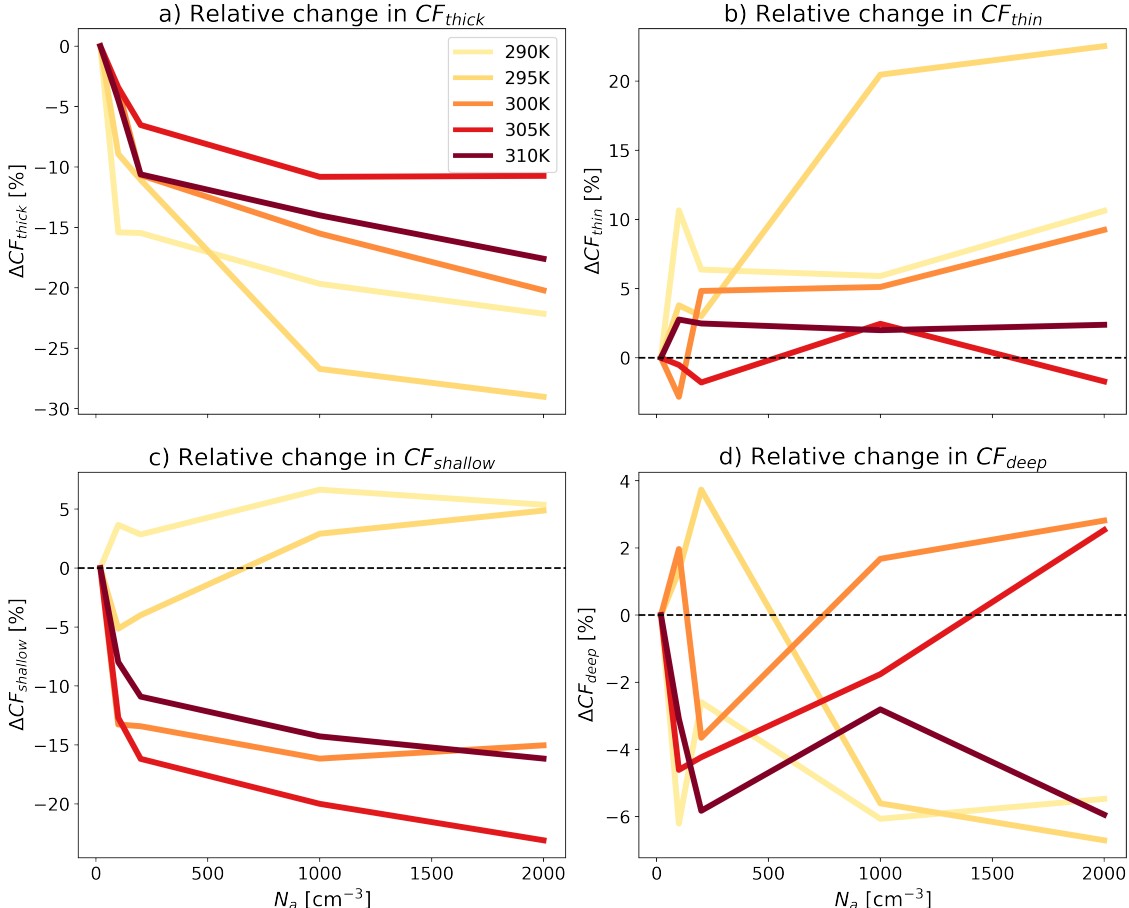

**Figure 5.** The relative response of domain and time mean cloud fraction of thick ice ($CF_{thick}$; **a**), thin ice ($CF_{thin}$; **b**), shallow ($CF_{shallow}$; **c**) and deep convective clouds ($CF_{deep}$; **d**) to an increase in $N_a$. The values are shown as a difference, relative to the cleanest run (as denoted by the $\Delta$ sign), for each SST. The baseline cloud fractions are presented in Fig. S8, SI.

$$CRE = \int_{0}^{\infty} \int_{0}^{\infty} CRE\left(\mathcal{L},\mathcal{I}\right) CF\left(\mathcal{L},\mathcal{I}\right) d\mathcal{L} d\mathcal{I} \qquad (4)$$

In the simulations presented here $\Delta CRE \cong \Delta R$ (Fig. 1). Thus, following a somewhat similar method to that presented in Bony et al. (2004) and Sokol et al. (2024), we decompose the mean $\Delta R$ into three contributions:

$$\Delta CRE \cong \Delta R$$

$$= \underbrace{\int\limits_0^\infty \int\limits_0^\infty \Delta CRE(\mathcal{L},\mathcal{I}) CF(\mathcal{L},\mathcal{I}) \, d\mathcal{L} \, d\mathcal{I}}_{Opacity}$$

$$+ \underbrace{\int\limits_0^\infty \int\limits_0^\infty CRE(\mathcal{L},\mathcal{I}) \Delta CF(\mathcal{L},\mathcal{I}) \, d\mathcal{L} \, d\mathcal{I}}_{Shift}$$

$$+ \underbrace{\int\limits_0^\infty \int\limits_0^\infty \Delta CRE(\mathcal{L},\mathcal{I}) \Delta CF(\mathcal{L},\mathcal{I}) \, d\mathcal{L} \, d\mathcal{I}}_{Nonlin} \tag{5}$$

In this decomposition, the first term on the right-hand-side, the "Opacity" term, represents changes in $\Delta R$ due to changes in the CRE per $\mathcal{L}$ and $\mathcal{I}$ bin, while the distribution of $\mathcal{L}/\mathcal{I}$ are held fixed, i.e., this term is calculated by multiplying Fig. 4a with Fig. 4k. This term represents changes in the cloud's opacity (reflectance and absorption) for a given liquid and ice amount (for example by the Twomey effect). We note that this term could also be influenced by changes in clear-sky fluxes (Sokol et al., 2024). The second term on the right-hand-side, the "Shift" term, represents changes in $\Delta R$ due to changes in the distribution of $\mathcal{L}/\mathcal{I}$ occurrence, while the CRE per $\mathcal{L}$ and $\mathcal{I}$ bin is held fixed, i.e., this term is calculated by multiplying Fig. 4b with Fig. 4l. The Shift term is contributed by both changes in the total CF and by a shift between the different cloud regimes (for example thinning of ice clouds). The last term on the right-hand-side, the nonlinear ("Nonlin") term, represents the combined effect of changes in the CRE and the cloud occurrence in the different $\mathcal{L}/\mathcal{I}$ bins, i.e., this term is calculated by multiplying Fig. 4k with Fig. 4l.

Fig. 6a-c illustrates the decomposition presented in Eq. 5 for the domain mean (i.e., integrating over all $\mathcal{L}$ and $\mathcal{I}$ bins, excluding the no clouds regime as defined in Table S1, SI) for all the different SSTs. Fig. 6a-c also present the simulated response as presented in Fig. 1 (referred to as "Model") and the sum over the three terms presented in Eq. 5 (referred to as "Total"). These panels illustrate that the Opacity term is the main driver for the decline in $\Delta R$ with $N_a$ (Fig. 6a), occurring mostly through the SW (Fig. 6b). In addition, this figure illustrates that the Opacity term is the main driver for the SST-sensitivity, demonstrating a generally weaker response as the SST increases, consistent with Fig. 1c. The Shift term, on the other hand, demonstrates similar magnitudes but opposite sign in the SW and LW (Figs. 6b and c, respectively), with a weak SST-dependence, thus making this term negligible in the total (Fig. 6a). The nonlinear term shows close to zero contributions to $\Delta R$ and its SW and LW components, thus justifying focusing on the linear decomposition in Eq. 5. We note that the decomposition results in a similar magnitude and SST-trend as the model (comparing Total to Model in Fig. 6a-c), thus justifying its use. However, we also note a slight over-estimation of $\Delta R$ using the decomposition at the lower SSTs (Fig. 6a).

In addition to the domain mean, $\Delta R$ is decomposed per cloud regime by integrating over the relevant part of the $\mathcal{L}$ and $\mathcal{I}$ phase-space (Fig. 6d-f and Table S1, SI). These panels illustrate that deep convective clouds have negligible contributions to

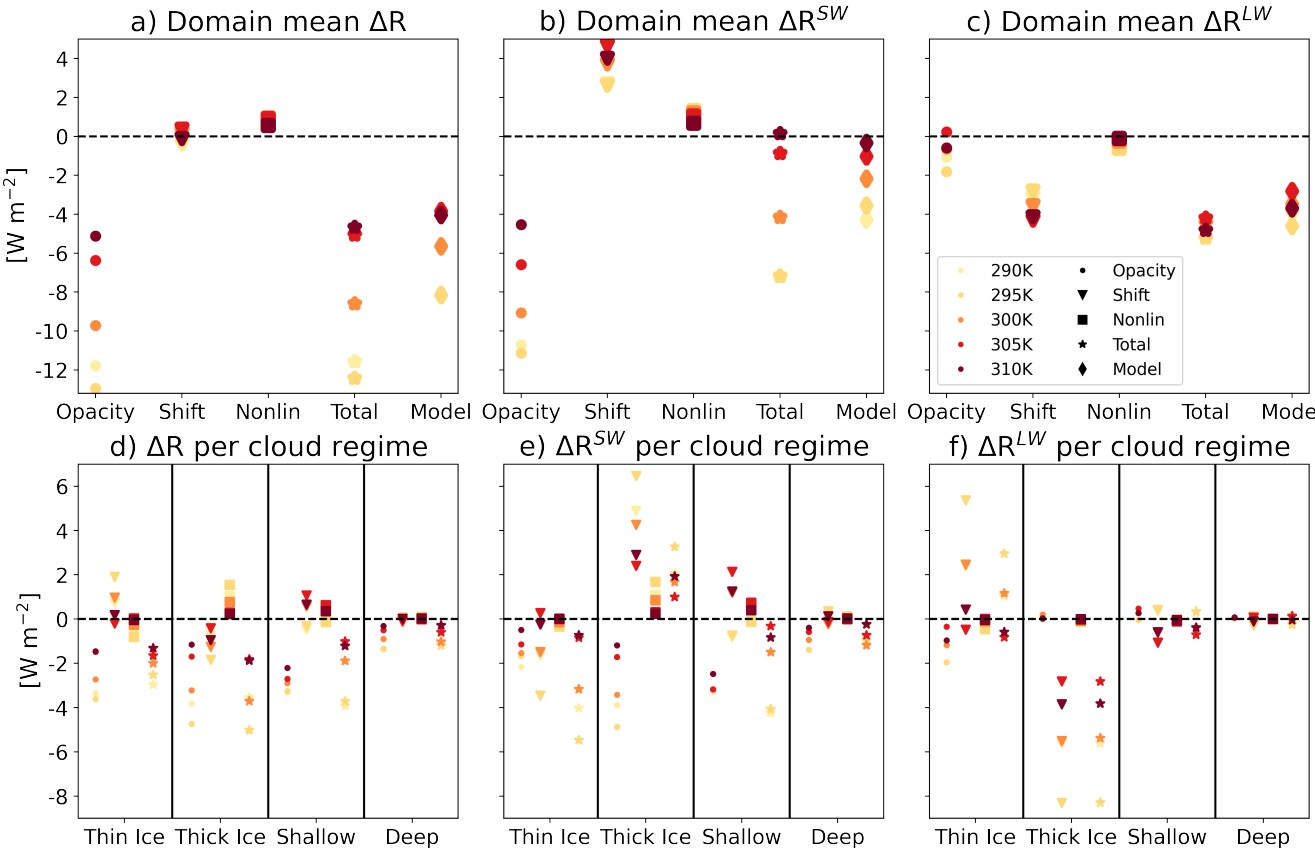

**Figure 6.** The time mean $\Delta R$, $\Delta R^{SW}$ and $\Delta R^{LW}$ due to an increase in $N_a$ for the domain mean (**a-c**), and per cloud regime (**d-f**). The values shown are decomposed to the three terms shown in Eq. 5 (Opacity, Shift and Nonlin), and the increase in $N_a$ is represented by the difference between the most polluted run ($N_a = 2000$ cm$^{-3}$) and the cleanest run ($N_a = 20$ cm$^{-3}$), for each SST.

$\Delta R$, $\Delta R^{SW}$ and $\Delta R^{LW}$, mostly due to their small coverage (Figs. 4a and S8d, SI). Therefore, most of the contribution to $\Delta R$
comes from anvil and shallow clouds changes.

The thick and thin ice clouds' response drives a negative net total $\Delta R$, which is stronger under lower SSTs (Fig. 6d). This trend is dominated by the Opacity term, which is driven almost entirely by the SW part of the spectrum (Fig. 6e). This term represents an increase in the reflectivity of the ice clouds for a given $\mathcal{L}$ and $\mathcal{I}$ distribution, and can be explained by a similar mechanism to the Twomey effect but for ice particles. We note that this result might differ under coupling of $N_a$ to
ice nucleating particles, which is not considered here. This term becomes stronger (more negative) with a reduction in SST, especially for thick clouds, due to an increase in the baseline CF of these clouds (Fig. S8, SI). The Shift term in thick ice clouds is strongly positive in the SW (Fig. 6e) and negative in the LW (Fig. 6f) due to the thinning of the ice clouds and the general reduction of the occurrence of these thick clouds (Figs. 4k and 5a). Thin ice clouds exhibit an opposite trend to thick clouds

in the Shift term, due to them increasing in CO with an increase in $N_a$ (Figs. 4k and 5b). However, the combined net effect of thick and thin ice clouds on the Shift term is low, due to them being similar in magnitude but opposite in sign (Fig. 6d).

Similarly to the ice clouds' response, the shallow clouds' response also drives a negative net total $\Delta R$, which becomes stronger under lower SSTs (Fig. 6d). As expected, changes in shallow clouds have a small impact in the LW (Fig. 6f), but a significant effect in the SW (Fig. 6e). As in ice clouds, the negative net total $\Delta R$ in the shallow clouds case is driven mostly by the Opacity term, which in this case can be explained by the classical Twomey effect. Unlike ice clouds, the Opacity term demonstrates a low sensitivity to the underlying SST, but the shallow clouds' Shift term exhibits strong SST-sensitivity. This term, while having a relatively small magnitude, is negative under low SSTs and positive under high SSTs, consistent with the relative change in $CF_{shallow}$, which is positive under low SSTs and negative under high SSTs (Fig. 5c). The contrasting response of $CF_{shallow}$ to $N_a$ under the different SSTs can be explained by warm rain inhibition at varying depths of warm layers. As was noted above, with an increase in SST, the warm layer depth increases, while an increase in $N_a$ acts to push warm rain formation to higher levels (Rosenfeld, 2000; Freud and Rosenfeld, 2012; Heikenfeld et al., 2019). Thus, under lower SSTs, for which the warm layer depth is relatively shallow, an increase in $N_a$ can inhibit warm rain (see Fig. 3g) and hence lead to an increase in $CF_{shallow}$. In contrast, under higher SSTs, for which the warm layer depth is relatively deep, an increase in $N_a$ drives warm rain inhibition at the lower levels, which is compensated for at higher levels of the warm section (Fig. 3g), thus eliminating the positive effect on $CF_{shallow}$.

The combined response of ice and shallow clouds to an increase in $N_a$, as explained in this section, can explain the reduction in $\Delta R^{LW}$ with $N_a$, the reduction in $\Delta R^{SW}$ with $N_a$ and its SST-sensitivity, and hence the reduction in $\Delta R$ with $N_a$ and its SST-sensitivity (Fig. 1).

### 3.3 Mechanism behind the ice cloud fraction's response to $N_a$

As was noted above, a decrease in $CF_{thick}$ with $N_a$ (Fig. 5a) leads to more outgoing LW radiation out of the atmosphere (Fig. 1b). In order to understand the reduction in $CF_{thick}$ and the ice cloud thinning with $N_a$, next we examine the sensitivity of the maximum (in the vertical dimension – see Fig. 7d) of the radiative-driven mass divergence (Bony et al., 2016, $D_r$) to $N_a$ under the different SSTs (Fig. 8). Fig. 8a illustrates that the $CF_{thick}$ is strongly correlated with $D_r$ (Pearson correlation coefficient $\approx$ 0.93 with P-value < 0.01). While the general decrease in $D_r$ with SST has previously been demonstrated (Bony et al., 2016), here we show that for a given SST, $D_r$ generally decreases with $N_a$ (Fig. 8b). The general reduction in $D_r$ with $N_a$ drives a general reduction in $CF_{thick}$ with $N_a$ for a given SST (Figs. 5a and 8c). This reduction in $D_r$ and $CF_{thick}$ with $N_a$ explains the reduction in $\mathcal{I}$ and cloud ice (Figs. 2b and 3h, respectively) with $N_a$, which in turn can explain the reduction in $\Delta R^{LW}$ (Fig. 1b).

In addition to modifying $D_r$, an increase in $N_a$ also affects the lifetime of anvil clouds by perturbing the sedimentation rate (Grabowski and Morrison, 2016). Specifically, high aerosol conditions lead to smaller ice crystals, which sediment slower from the cloud (i.e. the sedimentation flux becomes less negative; Fig. S13, SI), thus acting to increase $CF_{thick}$. However, Fig. 5a shows a decrease in $CF_{thick}$ with $N_a$ in our simulations, thus making this to be only a secondary effect compared with the effect of $D_r$ (agreeing with previous results regarding the effect of warming on anvil clouds (Beydoun et al., 2021)).

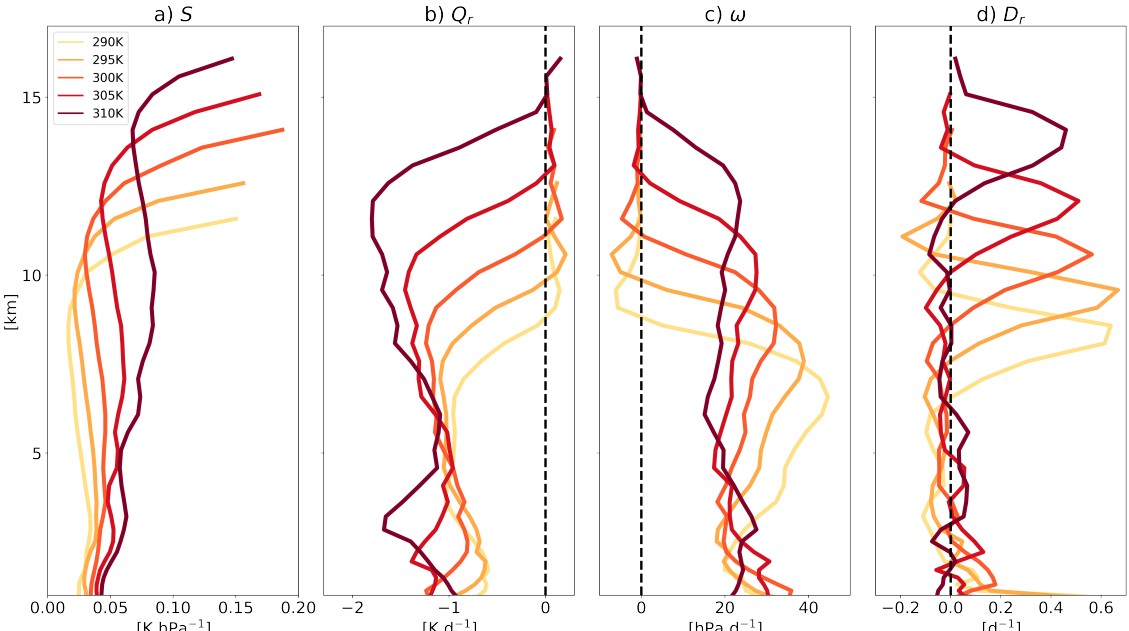

**Figure 7.** Domain and time mean vertical profiles of the: **a)** static-stability – $S$, **b)** radiative cooling rate – $Q_r$, **c)** vertical pressure velocity – $\omega$, and **d)** radiative-driven mass divergence – $D_r$ for the different simulations conducted under $N_a = 20$ cm$^{-3}$ and different SST conditions.

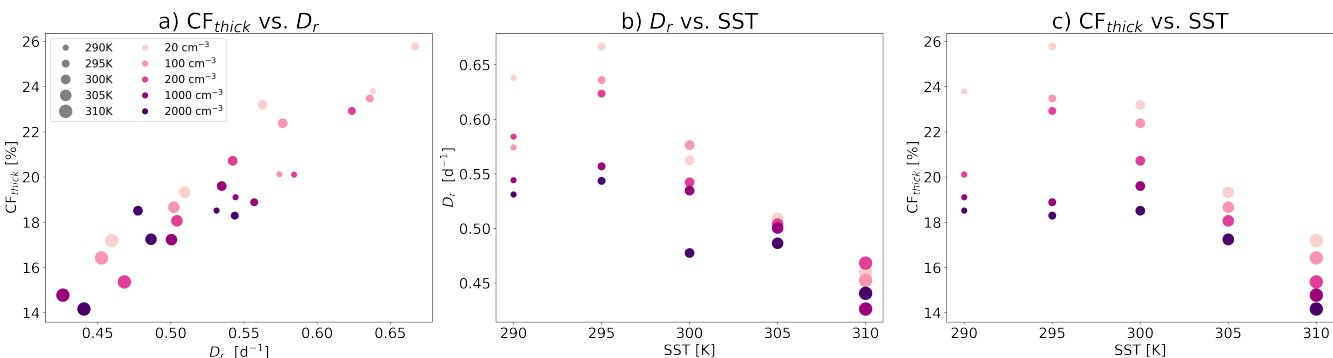

**Figure 8.** Changes in domain and time mean thick ice cloud fraction ($CF_{thick}$) with $D_r$ for the different simulations conducted under different $N_a$ and SST (**a**), changes in $D_r$ with SST (**b**), and changes in $CF_{thick}$ with SST (**c**).

A reduction in $D_r$ with $N_a$ could be attributed to changes in $Q_r$ (the radiative cooling rate; Fig. 7b) and/or in the static-stability ($S$; Fig. 7a). Thus, in order to understand the reasons behind the decrease in $D_r$ with $N_a$ (for a given SST), in Fig. 9 we calculate the change in $D_r$ with $N_a$ for the different SSTs, assuming that either $Q_r$ or $S$ are held fixed at the value it attains at a reference $N_a$ of 200 cm$^{-3}$ for each SST. This calculation is similar to that presented in Fig. 4 of Bony et al. (2016), but for changes in $N_a$ instead of changes in SST. Fig. 9 illustrates that the reduction of $D_r$ with an increase in $N_a$ can mostly be

attributed to changes in $S$. This result is illustrated by the consistent reduction in $D_r$ with $N_a$ for all SSTs when only $S$ (or the temperature – $T$) is varied. However, when only $Q_r$ is varied, the trend of $D_r$ with $N_a$ is not consistent across the different SSTs, and for some of the SSTs, the trend is not monotonic.

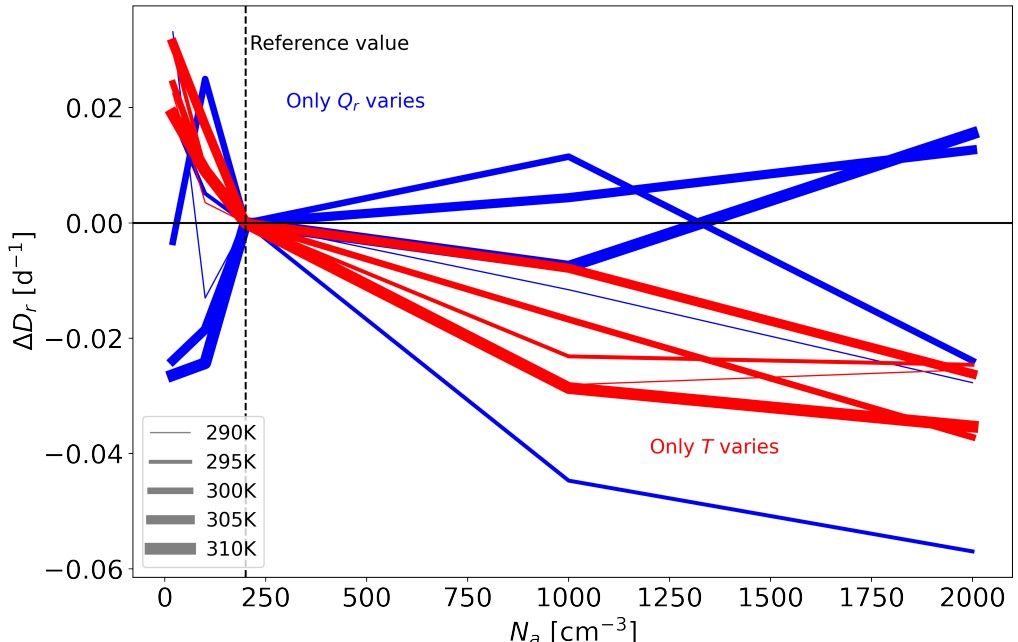

**Figure 9.** Relationship between the radiative-driven divergence ($D_r$) and $N_a$, diagnosed by assuming that only either the temperature profile ($T$ – red curves) or the clear-sky radiative cooling profile ($Q_r$ – blue curves) vary with $N_a$. The reference for the $T$ and $Q_r$ are the simulations conducted under $N_a = 200 \, \text{cm}^{-3}$ (dashed vertical line) for each SST.

The domain and time mean temperature vertical profiles for the different simulations and their response to an increase in $N_a$ is presented in Fig. 10. This figure illustrates that, for a given SST, an increase in $N_a$ drives strong warming of the upper troposphere, and in some cases a weak cooling of the lower troposphere. This trend demonstrates an increase in $S$ with $N_a$, which in turn explains the reduction in the anvil cloud fraction.

A remaining open question concerns the reasons behind the strong warming of the upper troposphere (or the increase in $S$) with $N_a$. In the model, a central prognostic variable is the liquid/ice water static energy ($h_L$). The $h_L$ tendency equation contains 5 terms: advection (adv), radiation (rad), latent heating (includes latent heating from freezing), turbulence and large-scale tendency (Khairoutdinov and Randall, 2003). In an RCE configuration, by definition, the large-scale tendency is set to zero, thus having no effect here. In addition, in our simulations the turbulence term is negligible compared to the rest of the terms. Hence, in Fig. 11 we present vertical profiles of the domain and time mean $\frac{\partial h_L}{\partial t}$ due to latent heating, advection,

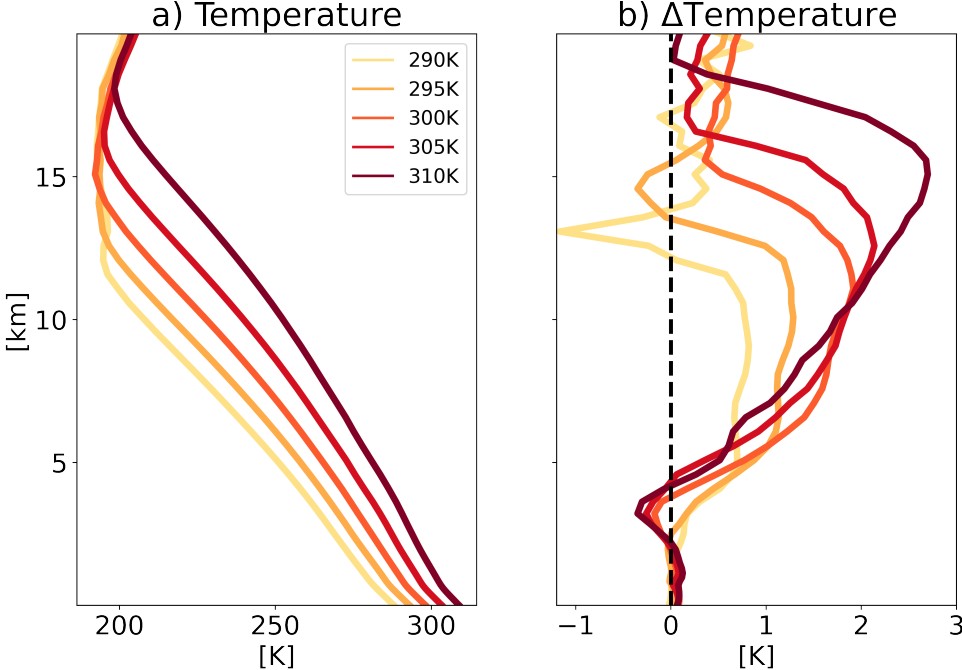

**Figure 10.** Domain and time mean vertical profiles of temperature of the cleanest runs ($N_a = 20$ cm$^{-3}$; **a**) and its response to increasing $N_a$ to 2000 cm$^{-3}$, relative to the cleanest run for each SST (**b**). Here we only present the cleanest and the response of the most polluted runs for clarity. The full range of $N_a$ is presented in Fig. S9, SI.

and radiation of the different simulations. This figure illustrates that under equilibrium conditions, the latent heating acts to heat the upper troposphere, advection acts to cool it, although by a smaller magnitude, and radiation acts to weakly cool the entire troposphere almost uniformly. This trend is enhanced with an increase in $N_a$ (Fig. 11d-f), suggesting that the increase in temperature of the upper troposphere with $N_a$ is mostly driven by a stronger latent heat release, which is consistent with the higher production rates of graupel and snow with $N_a$ (Figs. 3i and j). Graupel and snow, unlike small ice crystals, efficiently sediment out of the cold portion of the cloud, thus leaving behind the heat they released in their formation, resulting in a net warming effect. In addition, in higher altitudes, the air density drops. Thus, a given amount of latent heating will cause a larger temperature change at higher altitudes than low altitudes (Gasparini et al., 2023). Therefore, higher production of graupel and snow with $N_a$ is identified as the main driver of the observed temperature increase in the upper troposphere.

## 3.4 Examining the surface precipitation response to aerosol perturbation using the atmospheric energy budget

Next, we examine the response of the surface precipitation to $N_a$ under the different SSTs. Fig. 12a illustrates an increase in surface precipitation (in energy units – $L\Delta SP$, where $L$ is the latent heat of vaporization and $SP$ is the surface precipitation)

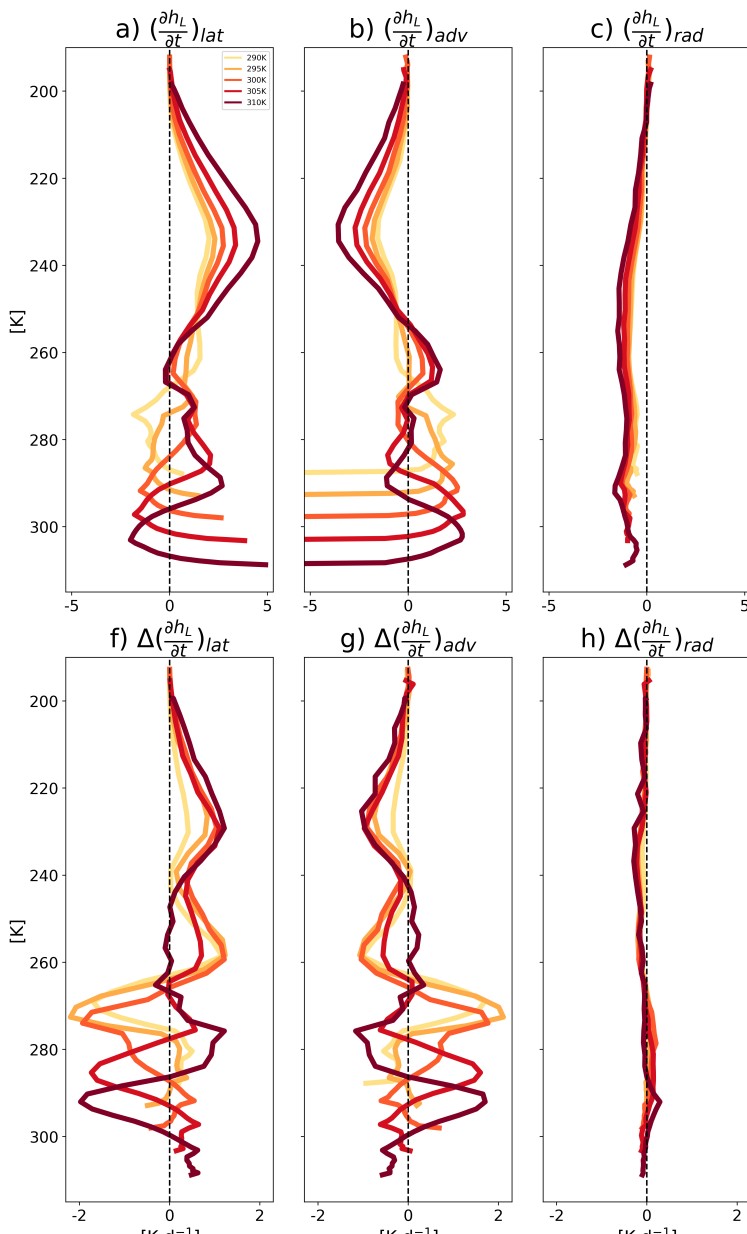

**Figure 11.** Vertical profiles of the domain and time mean tendency of the liquid/ice water static energy ($h_L$) for the cleanest runs ($N_a = 20$ cm$^{-3}$) due to (**a**) latent heating, (**b**) advection, and (**c**) radiation in the different simulations conducted under different SST and $N_a$. Panels **d** – **f** present the response of these terms to increasing $N_a$ to 2000 cm$^{-3}$, relative to the cleanest run for each SST. Here we only present the cleanest and the response of the most polluted runs for clarity. The full range of $N_a$ is presented in Figs. S10 - S12, SI.

with $N_a$ across SSTs. In order to understand this increase, we use the atmospheric energy budget perspective (Muller and O'Gorman, 2011; Dagan and Stier, 2020a; Williams et al., 2023) and decompose the changes in $L\Delta SP$ to changes in LW atmospheric radiative cooling ($\Delta LWC$, calculated as the TOA's LW radiation flux minus the surface's net LW radiation flux; Fig. 12b), changes in surface sensible heat flux ($\Delta SHF$; Fig. 12c), and changes in atmospheric SW absorption ($\Delta SWA$, calculated as the TOA's net SW radiation flux minus the surface's net SW radiation flux; Fig. 12d), following the notations of Williams et al. (2023):

$$L\Delta SP = \Delta LWC - \Delta SWA + \Delta SHF \tag{6}$$

We note that Eq. 6 holds under equilibrium conditions, as simulated here (Muller and O'Gorman, 2011; Dagan and Stier, 2020a). Following the notations of Eq. 6, Fig. 12a can be reconstructed by summing Fig. 12b-d. Hence, we note that the increase in $L\Delta SP$ could mostly be explained by enhanced $\Delta LWC$ (Fig. 12b), while changes in $\Delta SWA$ produce only a small positive contribution, and changes in $\Delta SHF$ present a small and non-consistent across SSTs contribution. The enhanced $\Delta LWC$ with $N_a$ is driven by clear-sky radiative cooling, which is in turn driven by the decreased $CF_{thick}$ with $N_a$ across SSTs, as illustrated in Fig. 5a. The enhanced $\Delta LWC$ is also consistent with the reduction in $\Delta R^{LW}$ presented in Fig. 1b. These results suggest that under equilibrium conditions, higher $N_a$ concentrations drive higher LW cooling rates of the atmospheric column, which supports the production of more precipitation.

## 4 Conclusions

Under anthropogenic-driven climate change, Earth's energy budget is influenced by changes in the atmospheric composition, including anthropogenic aerosols, which could affect the cloud radiative properties. In addition, changes in SST could drive changes in the cloud radiative properties as well, which can in turn further change the SST. In this study, we investigate the combined impact of SST and aerosol concentration ($N_a$) on cloud properties in the framework of high-resolution radiative-convective-equilibrium (RCE) simulations.

Using these idealized RCE simulations, we demonstrate that increasing $N_a$, which does not directly interact with radiation here, decreases top-of-atmosphere (TOA) energy gain across all SSTs, both in the longwave (LW) and shortwave (SW) parts of the spectrum, as a result of changes in the cloud radiative effect. We also show that this effect is stronger under lower SSTs, mostly in the SW, which is consistent with the stronger increase in liquid water path ($\mathcal{L}$) with $N_a$ under lower SSTs. On the other hand, the TOA outgoing LW radiation increases similarly across SSTs, consistent with a decrease in ice water path ($\mathcal{I}$). Lastly, the cloud fraction (CF) response to an increase in $N_a$ is negative, although not-monotonic and not-SST consistent.

To better understand these trends, we decompose the response of TOA energy gain ($\Delta R$) to different cloud regimes (based on 2D histograms of $\mathcal{L}$ and $\mathcal{I}$) and to contributions from changes in the cloud opacity (the Opacity term) and in cloud occurrence (the Shift term) based on a linear decomposition. This decomposition illustrates that most of $\Delta R$'s negative trend and its SST-sensitivity is driven by the Opacity term, which in turn is driven by the SW part of the spectrum. This trend can be explained by the Twomey effect, i.e, for a given $\mathcal{L}$ and $\mathcal{I}$ the clouds become more reflective with a rise in $N_a$. The Twomey effect is

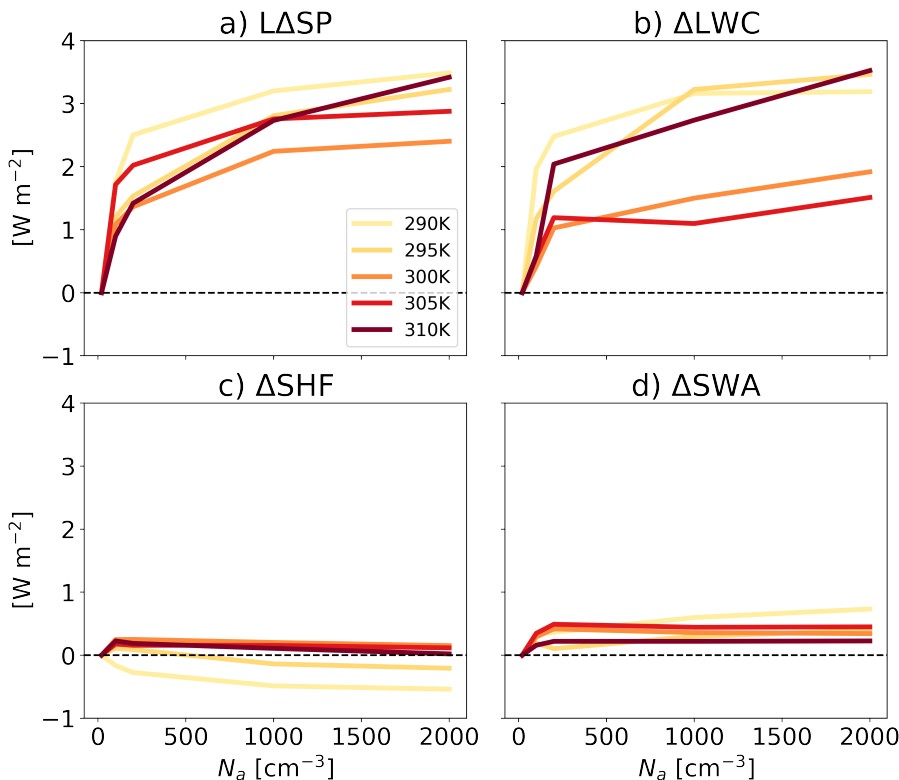

**Figure 12.** The response of domain and time mean surface precipitation ($L\Delta SP$; **a**), longwave atmospheric radiative cooling ($\Delta LWC$; **b**), surface sensible heat flux ($\Delta SHF$; **c**) and atmospheric shortwave absorption ($\Delta SWA$; **d**) to an increase in $N_a$, relative to the cleanest run for each SST ($N_a = 20$ cm$^{-3}$).

proportional to the baseline CF, thus becoming stronger under lower SST for which the baseline CF is higher. The Shift term,
on the other hand, illustrates a compensation between a positive response in the SW and a negative response in the LW, thus producing a small net effect. Furthermore, we decompose $\Delta R$ and its components per cloud regime, which illustrates that ice and shallow clouds are the main drivers behind the Opacity and Shift terms trends. Lastly, this cloud regime decomposition illustrates that, together with the general reduction in CF and specifically in thick ice cloud fraction ($CF_{thick}$), an increase in $N_a$ leads to the thinning of the ice clouds.

As has been previously reported (Bony et al., 2016), we observe a strong correlation between the ice CF, and specifically $CF_{thick}$, and the maximum radiative-driven mass divergence at the upper troposphere ($D_r$). We demonstrate that $D_r$ generally decreases with $N_a$ for a given SST, consistent with the reduction in $CF_{thick}$ and the shift of the anvil clouds toward thinner clouds (Fig. 4). The reduction in $D_r$ with an increase in $N_a$ is shown here to be driven by an increase in static-stability at the upper troposphere under more polluted conditions (Fig. 9). The decrease in anvil cloud fraction with $N_a$ across SSTs also leads
to a decline in $\mathcal{I}$, causing an increase in the outgoing LW radiation, i.e., decreasing $\Delta R^{LW}$. This reduction in $\Delta R^{LW}$ at the

TOA directly increases LW cooling of the atmospheric column ($\Delta LWC$), which, in turn, is identified as the main driver of enhanced surface precipitation ($L\Delta SP$). We note that an increased surface precipitation could mean that aerosols get rained out faster, thus moderating the aerosol concentration. In our simulations $N_a$ is prescribed, thus this feedback is disabled. This feedback should be examined in future studies.

Lastly, we try to explain the observed relative warming of the upper troposphere with $N_a$, which is consistent with the rise in static-stability, by examining the tendency equation of liquid/ice water static energy ($\frac{dh_L}{dt}$). We demonstrate that the increase in static-stability with $N_a$ can be explained by an increase in the latent-heating of the upper troposphere. Warm rain inhibition with $N_a$ leads to heightened production rates of graupel and snow, which efficiently sediment out from the colder region of the cloud. As they descend, they leave behind the latent heat released during their formation, resulting in an overall warming effect and an increased stability.

The results presented here are based on idealized RCE simulations in a small domain, which suppress convective self-aggregation and large-scale circulation. In a larger domain, the circulation is suggested to intensify with an increase in $N_a$ (Dagan, 2024; Dagan et al., 2023). In this case, the large-scale circulation changes dominate the change in the domain-mean cloud and radiative properties. In our simulations we are focusing on the local response and these larger-scale effects are not accounted for. Furthermore, the role of other modeling choices, such as horizontal and vertical resolution and the role of boundary conditions (Dagan et al., 2022) in our results should be examined in future work. In addition, in this work we excluded aerosol-radiation interactions, which could drastically alter TOA energy gain (Bellouin et al., 2020; Williams et al., 2023), and as such could be of interest. Finally, our work is based on single-model simulations. An RCEMIP stage focusing on aerosol effect on clouds and RCE climate is currently being conducted. This set of multi-model simulations under harmonized setup will allow us to confront our conclusions with a large variety of models and microphysical schemes.

This work suggests that under equilibrium conditions, the magnitude of the effective radiative forcing by aerosol-cloud interactions decreases (becomes less negative) with an increase in SST. These results predict that under the ongoing global warming trend, the ability of aerosol-cloud-interactions to counteract some of the positive radiative forcing by greenhouse gasses will become smaller with time. In addition, it suggests that studying the sensitivity of clouds to aerosol and SST should be conducted concomitantly as mutual effects are expected.

*Code and data availability.* SAM is publicly available at: http://rossby.msrc.sunysb.edu/ marat/SAM.html The data presented in this study is publicly available at: https://doi.org/10.5281/zenodo.8338310

*Author contributions.* SL carried out the simulations and analyses presented. GD assisted with the simulations. SL and GD designed and interpreted the analyses. SL prepared the manuscript with contributions from GD.

*Competing interests.*  At least one of the (co-)authors is a member of the editorial board of Atmospheric Chemistry and Physics.

*Acknowledgements.*  This research was supported by the Israel Science Foundation (grant number: 1419/21).

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
