# Peer review of "On the sensitivity of aerosol-cloud interactions to changes in sea surface temperature in radiative-convective equilibrium"

_EGUsphere, 2023_

## Referee Comment (RC2)

This is an interesting and useful study that examines how the assumed aerosol concentration $N_a$ impacts cloud characteristics for a variety of sea surface temperatures (SST's) in idealized radiative-convective equilibrium simulations. This study builds on previous studies that focused on either the SST effect on clouds or the aerosol effect on clouds, but not the combination of both effects. It is shown that increased aerosol concentrations enable liquid cloud droplets to loft higher in the atmosphere as they do not grow and rain out as quickly. For the cooler simulations, which have shallower warm layers, dropets are lofted above the freezing level before the droplets are large enough to form rain. This decreases rain rates but increases ice crystal growth and therefore frozen precipitation production. For the warmer simulations, thicker warm layers allow rain to still form, although at higher elevations, and so the precipitation phase is not as sensitive to $N_a$. This finding was made possible because the study considered both a range of both SST's and values of $N_a$, rather than one or the other, which illustrates the novelty of the study.

Interpretation of some of the results is problematic, which I have outlined further below. In particular, a central result of the study was that anvil cloud fraction is reduced with increasing $N_a$, but the authors do not clearly state how anvil cloud fraction is calculated and two plots showing anvil cloud fraction are inconsistent with each other. Furthermore, one plot appears to show anvil *height* changes with increasing $N_a$, but anvil *coverage* does not. Both observations call the anvil result into question.

The authors may need to revisit their analysis of simulation output in order to correct the inconsistency in the anvil cloud fraction. If the results change, the discussion and conclusions may need to be significantly revised. There are also some smaller claims that appear to be unsupported in plots, outlined below, which also necessitate a revision of some of the results/discussion and conclusion sections.

Once the issues with anvil cloud fraction are addressed, I believe there is significant potential for novel results about anvils that would attract a wide audience. In particular, it is possible that both FAT and the stability iris hypothesis are supported for a given $N_a$, but FAT does not apply across different $N_a$. Rather, increased $N_a$ raises and thins anvils. These guesses are based off a tenuous reading of the plots and would need to be examined further.

**General comments**

1. The claim that increasing $N_a$ decreases anvil cloud fraction (CF) is first made a line 157, in reference to Fig. 2c. Fig. 2a shows a maximum in CF at about 220K, and Fig. 2c shows increases in CF above this level and decreases below with increasing $N_a$. This indicates a rising of the level of maximum CF. The increase in CF above 220K is smaller in magnitude than the decrease in CF between 250K and 220K, which indicates the anvil depth decreases with an increase in $N_a$. Furthermore, if the zero-crossing in plot 2c is precisely at or below the maximum in plot 2a, as it appears, the value of the maximum CF could not be decreasing. However, it is difficult to tell in the plots.

   Anvil CF is further investigated at line 161 and Fig. 6a, which shows a relationship between anvil CF and maximum mass divergence. The plots are inconsistent given that all points in Fig. 6 have anvil CF< 0.24 but that all profiles in Fig. 2a have a maximum CF> 0.35 at about 220K. The authors need to describe carefully how anvil CF is calculated. My guess is that, for Fig. 6, anvil cloud fraction is calculated as the time mean of

$$\frac{\text{number of cloudy grid points above 245K}}{\text{total number of grid points above 245K}}$$

   This is not the anvil cloud fraction, but rather the anvil cloud volume. If true, this would explain why Fig. 6a shows a decrease in anvil CF, but Fig 2 does not: anvil depth decreases, decreasing their volume, but not necessarily their horizontal extent. The authors should calculate anvil cloud fraction as the maximum in the vertical profile of cloud fraction above 245K, which would be the same definition as Beydoun et al., 2021 and similar to the definition in Saint-Lu et al., 2020.

2. In multiple locations, a distinction is made between the cloud radiative effect specifically and the overall changes to radiative effects generally. But aerosols were only allowed to change cloud microphysics and

not interact with radiation directly, and thus the CRE was effectively the only thing that was allowed to change in the simulations. Indeed, the top and bottom rows in Fig. 1 appear nearly identical.

Specifically, the following statements are misleading and should be reworded:

(a) L5: "we show that increasing $N_a$ leads to a decline in top-of-atmosphere (TOA) energy gain across SSTs due to changes in the cloud radiative effect"

(b) L130: "the CRE ... is identified as the main driver of $\Delta R$ variations, while changes in clear sky radiation has a minimal impact"

(c) L218: "increasing $N_a$ decreases top-of-atmosphere (TOA) energy gain ... as a result of changes in the cloud radiative effect."

In all cases, I suggest it be made clear that the aerosol concentration affects the CRE, which is the only component of the overall aerosol radiative effect that is examined.

3. The introduction presents a nice overview of research on how clouds respond to changes in SST, including the FAT hypothesis and the stability iris effect. There is also a nice overview of existing work on how aerosols affect cloud properties. However, while there is a clear motivation for investigating higher SST's, the reader may be left wondering how realistic the aerosol concentration scenarios used are, if and how global aerosol concentrations are expected to change under warming, and how the results of the study would modify the aerosol concentrations further.

For example, the study finds increased aerosol concentrations may increase precipitation rates. One may hypothesize that such increased precipitation would feed back into the aerosol concentration as the aerosols get rained out faster, and thus moderate the concentration. Even if these effects are highly uncertain, they could be discussed.

4. The supplement presents plots in the same format as the plots in the main text, but every plot has all combinations of SST and $N_a$. Including all 25 profiles on each plot makes them impossible to read. I would create a new plot for each value of $N_a$ so that these plots can be read. The plots would be exactly as the top row (or only row) in Figs. 1,3,5,8,9, but for different $N_a$.

5. Most vertical profiles in the study are plotted with temperature as the vertical coordinate. I find this quite useful in some cases, e.g. fig 2a appears to support the FAT hypothesis. In other cases, it is slightly more confusing, e.g. in determining the altitude of rain production in figs. 3b and 3g.

The only change I would request here is that there appears to be an artifact in most profiles that could be removed. For example, the cloud ice profile (Fig 3c) is bivalued between $\sim$ 200k and $\sim$ 230K. I assume one value is from the stratosphere while the other is from the troposphere. I suggest truncating the profiles where the temperature begins to increase, thus ensuring each $T$ is uniquely associated with a single $z$.

**Specific comments**

1. L5: ", even at equilibrium conditions.": Things not at equilibrium include the SST with insolation and $N_a$ with aerosol removal via deposition. I suggest explicitly stating the equilibrium is RCE.

2. L133: Please describe how grid points are defined as cloudy for the purpose of calculating cloud fraction.

3. L135: Does "total water" refer only to total liquid and ice? If so, please clarify. "total water" to me reads as including water vapor.

4. L146: " under lower SST ... an increase in $N_a$ can completely suppress warm rain (see Fig. 3g)." This appears unsupported in the plot. The cleanest 290K run shows a maximum for rain of $\sim 0.004$g kg$^{-1}$, while the change in rain in the polluted scenario shows a maximum change of $\sim -0.002$g kg$^{-1}$. Please quantify this statement.

5. L151: "The stronger increase in total water with Na under lower SSTs leads to the stronger SW reflectivity,": I'm not sure that just the result of increased total water is enough to establish causality here, with the role of anvils and changes to cloud morphologies uncertain. Perhaps just "consistent with".

6. L156: "higher $N_a$ and lower SSTs ... thus producing more graupel (Fig. 3i)" In Fig. 3i, the coldest run shows the smallest increase in graupel production. The largest increase is from the middle 300K simulation, so there is not a clear trend.

7. L169: "A reduction in $D_r$ with $N_a$ could be attributed to changes in $Q_r$": For me, it's less intuitive that $N_a$ would affect $Q_r$ than $D_r$. Throughout this section, it is not clear to me that causality can be estabished, though I don't understand the calculation. Please explain the calculation and consider whether it establishes causality.

8. L178: "for a given SST, an increase in Na drives strong warming of the upper troposphere and a weak cooling of the lower troposphere.": The two coldest simulations do not appear to cool in the lower troposphere. They warm in the upper troposphere, but both have small regions of cooling in the upper troposphere.

9. L179: "the increase in S with Na (Fig. 5a)": Fig. 5a does not show the change with $N_a$, only the values for the cleanest simulation.

10. L190: "driven by a stronger latent heat release,": As well as weaker advection (since the advection term is negative), which appears to be a similar magnitude.

11. L190: "is in turn driven by higher production rates of graupel and snow ": I would say "consistent with". The causality implied here is hard to see and may flow the other direction.

12. L199: Kirchoff's Law implies $\Delta LWC \approx -\Delta R^{LW}$ because net surface LW radiation flux is zero if LW surface reflection is negligible. Is this true? The plots of $\Delta LWC$ and $-\Delta R^{LW}$ appear almost identical.

13. L219: "lower SSTs, contributed mostly by a stronger SW response.": Fig. 1 shows, especially in the colder simulations, quite similar SW and LW components to the CRE.

14. L228: I'm not sure this is shown clearly enough to be a conclusion. Cloud morphology, overlap, and anvil coverage may also impact the SW reflection.

15. L245: This paragraph generally nicely explains the limitations of the simulation. I would perhaps also add that the finding that increased $N_a$ increases precipitation could lead to moderation of $N_a$ through faster rates of wet deposition, but that here $N_a$ was prescribed.

Figures:

1. Fig. 7: Please explain how this is calculated. It is not obvious to me how $Q_r$ could be held fixed in any given simulation.

**Minor comments**

1. L74: "vertical pressure velocity" $\rightarrow$ "clear sky vertical pressure velocity"

2. L81: "and observations to decrease with surface temperature " $\rightarrow$ "and observations to decrease in coverage with surface temperature "

3. L112: "solar-insulation" $\rightarrow$ "solar insolation"

4. L115: "is fixed at pre-industrial level (280 ppm)" $\rightarrow$ "is fixed at the pre-industrial level (280 ppm)"

5. LL118: "The vertical profile of $O_3$'s represents": I suggest simply $O_3$ or just "ozone"

6. L141: "the isotherm height" $\rightarrow$ "freezing level"

7. L160: " reduction in cloud ice at the upper troposphere (Fig. 3f),": I think you meant 3h.

8. L199: "calculated as the TOA's LW radiation flux" → "calculated as the TOA's net LW radiation flux". Right?

9. L227: "Hence, the liquid water content in clouds" → "Hence, the column liquid water content in clouds"

10. L256: "ACI": Spell out explicitly here please.

Figures:

1. Fig. 3: Some x-axis labels are overlapping and difficult to read.

2. Fig. 4: "and ice water path (IWP; a)" → "and ice water path (IWP; b)"

3. Fig. 5: "c) vertical pressure velocity" → "c) clear sky vertical pressure velocity"

**Suggestions**

The comments in this section are intended to be more suggestive. The authors are welcome to take them into consideration but need not feel obligated to.

1. L80: "Beside increasing in height": This is the first explicit mention of the prediction that anvils will rise with FAT. If a reader was not familiar with FAT, it could be nice to introduce it in the paragraph beginning at line 67.

2. Fig. 4 and 10: Perhaps my personal preference, but I would plot on the y-axis the values rather than the difference with the cleanest run. The point would remain the same as this amounts to a vertical shift. If you choose not to, make this more clear in the captions ("relative to the cleanest run for each SST" → "Values are shown as a difference with the cleanest run for each SST." or similar)

3. I might add a thin dashed line on all plots that present changes along the $\Delta\phi = 0$ line where $\phi$ is the variable being plotted (similar to Fig. 10).

4. The font sizes in plots are inconsistent. Assuming these are generated in Python, this could be fixed by setting `figsize` to a constant value in `plt.subplots(figsize=(width, height))` and then setting the font sizes to constant values in `ax.set_ylabel()`, `ax.set_title()` etc.

5. L182: Perhaps define $h_L$ mathematically to make clear this includes latent energy from freezing.

6. L220: "pushes warm rain initiation to higher levels of the cloud" → "pushes would-be warm rain initiation to higher levels of the cloud" or something similar. In the colder simulations, these "higher levels" are above the freezing level, so rain does not form as well since the ice crystals "steal" water off the droplets that would become rain. In Fig. 3g, it is only the warmest 2 simulations that show increased rain near the freezing level. This is explained well in the following sentences.

7. L253: "allow us to confront our conclusions": Did you mean confirm?

**References**

Beydoun, H., Caldwell, P. M., Hannah, W. M., & Donahue, A. S. (2021). Dissecting anvil cloud response to sea surface warming [e2021GL094049 2021GL094049]. *Geophysical Research Letters*, *48*(15), e2021GL094049. https://doi.org/https://doi.org/10.1029/2021GL094049

Saint-Lu, M., Bony, S., & Dufresne, J.-L. (2020). Observational evidence for a stability iris effect in the tropics [e2020GL089059 10.1029/2020GL089059]. *Geophysical Research Letters*, *47*(14), e2020GL089059. https://doi.org/https://doi.org/10.1029/2020GL089059

---

## Author Comment (AC1)

**Reply to the referees' review of:**

**On the sensitivity of aerosol-cloud interactions to changes in sea surface temperature in radiative-convective equilibrium**

**Reviewer #1**

We would like to thank Reviewer #1  for his constructive suggestions. Please find below a point-by-point reply to all the referee's comments (in blue).

Lorian and Dagan, 2023 uses idealized simulations in radiative-convective equilibrium to study the effects of changing aerosol concentrations on radiation and precipitation at different sea surface temperatures. The authors find that while the effects on radiation are qualitatively similar at all temperatures, the effect decreases with increasing SST. The manuscript attributes the decrease in SST-mediated changes in aerosol-cloud interaction to differential aerosol effects on warm rain formation and cloud liquid in lower parts of deep convective clouds. These become deeper at higher SSTs, leading to different responses. Another interesting result I want to highlight here is the anvil cloud fraction and ice water path decrease with increasing aerosol number, that is caused by increased latent heating and consequently increased upper tropospheric stability, following the "stability iris" hypothesis.

This manuscript presents several novel results on a very important and understudied topic. However, I am not convinced about some of the proposed explanations. In particular, I remain skeptical about the radiative relevance of changes in the warm parts of deep convective clouds. Can these really modify changes in radiative fluxes at the top of the atmosphere, given that they lie below very reflective thick anvil clouds? Furthermore, my main suggestion to the authors would be to go beyond the domain-average perspective and try to decompose changes in different cloud types (for example active deep convective cores, thick anvils, thin anvils, low clouds).

Overall, I found the manuscript to be an interesting read, full of noteworthy insights. However, I would strongly encourage that the authors to consider further clarification or even revision of some of their key findings to improve the overall quality of their work prior to final publication. For a more detailed discussion, please see my extended comments below.

**General comments:**

We reply to the next two comments together as they are strongly connected.

1. Regions dominated by deep convective lifecycle that are at/near conditions of radiative-convective equilibrium are typically dominated by anvil clouds, both in terms of coverage as well as radiative impact (e.g. Berry and Mace, 2014). The radiative importance of deep convective cores is, despite their strong LW and SW radiative effects, relatively small due to their small coverage. Changes in their properties/frequency are therefore unlikely to substantially modify climatological CRE. Moreover, given the large IWP of deep convective cores (see e.g. Fig. 1 in Sokol and Hartmann, 2020), I find it hard to believe that changes in the warm parts of deep convective clouds can directly influence TOA radiation. Couldn't changes in TOA radiative fluxes be entirely related to changes in anvil clouds?

2. On the other hand, even in disaggregated RCE simulations, there is a small but non-negligible population of stratiform low clouds as shown in Figure 1a. What fraction of changes in cloud liquid, rain, and their radiative implications can be attributed to changes in stratiform liquid clouds?

→As mentioned above, it is difficult to answer my two key questions without digging deeper into the model output, beyond the domain averages. My first thought would be to perform an analysis of the 2D SAM model output fields and subdivide the domain into, for example, areas of active deep convection, thick anvils, thin anvils, and low clouds. This could be done, for example, based on the diagnosed ice water path fields. In addition, it may be useful to look at changes in COD/in-cloud CRE to distinguish between effects on cloud fraction and cloud opacity.

**Reply:** Many thanks for raising this important point and for the nice suggestion to examine the response by cloud type. Following this comment significant modifications were implemented into the revised manuscript as elaborated below.

For examining the different cloud types, in the revised manuscript we are presenting 2D histograms of liquid water path ($\mathcal{L}$) and ice water path ($\mathcal{I}$) and the response of these cloud types to changes in $N_a$. The figure below, which was added to the revised manuscript as Fig. 4, presents the 2D histograms of the cloud fraction (CF) at the different bins of $\mathcal{L}$ and $\mathcal{I}$ and the average total, shortwave and longwave cloud radiative effect (CRE) at these different bins. These quantities are presented for two simulations using the lowest ($20 \ \mathrm{cm}^{-3}$; panels a-d) and the highest ($2000 \ \mathrm{cm}^{-3}$; panels e-h) $N_a$ and SST = 290 K. In addition, the difference between the high and low $N_a$ conditions is presented in panels i-l. As the reviewer mentioned, this figure illustrates that the CF in these RCE simulations is dominated by anvil clouds (e.g. Wing et al., 2020), i.e., clouds with negligible $\mathcal{L}$ and high (thick anvil clouds; denoted by marker 1 in Fig. 4a) or low (thin anvil clouds; denoted by marker 2 in Fig. 4a) $\mathcal{I}$. However, this figure also demonstrates the existence of two+ other types of clouds in these RCE simulations - shallow clouds (high $\mathcal{L}$ and low $\mathcal{I}$; denoted by marker 3 in Fig. 4a) and deep convective cores (high $\mathcal{L}$ and high $\mathcal{I}$; denoted by marker 4 in Fig. 4a). Examining the difference between the high and low $N_a$ simulations demonstrates that an increase in $N_a$ drives a thinning of the anvil clouds, i.e., an increase in the frequency of thin anvil clouds and a decrease in the frequency of thick anvil clouds (Fig. 4i). It also demonstrates that with the increase in $N_a$ the CRE becomes more negative for all $\mathcal{L}$ and $\square$ (and especially for low $\mathcal{I}$ and medium-high $\mathcal{L}$; Fig. 4j) bins, driven mostly by changes in the SW (Fig. 4k), which are driven by the Twomey effect (Twomey, 1977), with only a minor change in the LW (Fig. 4l). This interesting trend is now presented and discussed in the revised manuscript, which we believe improved the contribution of our paper.

In addition to the continuum/histogram perspective presented in Fig. 4, we examine the relative change in the total ice CF ($\boxed{\phantom{xxx}}$) with $\square$, which is calculated by integrating over thick ice plus thin ice regimes as defined in Table S1, SI (see below; see also Sokol et al., 2024). This calculation demonstrates that, as was reported in the original manuscript, the $CF_{ice}$ is reduced with an increase in $N_a$ for all SSTs examined here (Fig. 5a). In addition to the changes in the $\boxed{\phantom{xx}}$, as the reviewer suggests, there is a non-negligible shallow cloud fraction ($\boxed{\phantom{xxx}}$, calculated as the integral over the shallow clouds regime as defined in Table S1, SI) in the domain, which affects the SW (Fig. 4c). These shallow clouds become more reflective in the SW with an increase in $\square$ due to the Twomey

effect (Fig. 4k), and their fraction also changes in a way that depends on the SST (Fig. 5b). Specifically, Fig. 5b demonstrates that ☐☐☐☐ increases with ☐ under low SSTs and decreases with ☐ under high SST. This trend can help understand the higher SW sensitivity to $N_a$ under low SSTs compared with high SST, as under high SST the Twomey and the $CF_{shallow}$ effects counteract each other while under low SSTs they add to each other. Please see more details about this decomposition below.

In addition, following the reviewer's suggestion, we are decomposing the radiative response to contributions from changes in CF and from changes in cloud optical depth.This is done by a linear decomposition following Bony (2004) and Sokol (2024). This analysis demonstrates that the net radiative response is coming from changes in both ice and shallow clouds. Please see the details below.

The additions to the revised manuscript:

[revised manuscript text omitted]

1. Figures 3a and 6h make me wonder if the effect of aerosols on clouds also affects cloud height, cloud top temperature, and thus the radiative budget. High aerosol simulations seem to lead to higher, colder clouds. This may be a radiatively important mechanism that should be discussed. Is the decrease in cloud top temperature strong enough to significantly affect the LW CRE?

**Reply:** Thank you for raising this important question. Following this comment we examine the sensitivity of the cloud top temperature and height to ☐. For doing that, we define the domain-mean cloud top to be at the level for which the domain-mean total condensed water crosses an arbitrary threshold. Three different thresholds were used (☐) in order to check if the results are threshold dependent. For each SST, we used the cleanest run (☐ = 20 ☐) as a reference. As can be seen in the figure below (Fig .R1), cloud top height changes with aerosols are non-monotonic, non SST- and threshold-consistent and small (relative to the vertical resolution, i.e., the difference is only one grid point to each direction - Fig. R1b, d and f), for all three thresholds used here. For a given cloud top height, the cloud top temperature does generally slightly increase with ☐ due to warming of the upper troposphere (Fig. 10 in the main text). However, here again the change is non-monotonic and non SST- and threshold-consistent (Fig R1a, c and e). Thus, we conclude that changes in the cloud top temperature and height with aerosols do not significantly affect the radiation budget.

[Figure]

**Figure R1.** The response of domain and time mean cloud top temperature (left column) and cloud top height (right column) to an increase in ☐, both for three threshold choices: [          ] (first row), [          ] (second row) and [          ] (third row). The values are presented relative to the cleanest run ( ☐ = 20 ☐ ) for each SST, as indicated by the ☐ sign.

2. While this is clearly an idealized study, I am not convinced that using the $N_a$ of 20/cc is the best choice. To my knowledge, the $N_a$ should be higher even under very clean conditions.

**Reply:** Thank you. Indeed, this is an idealized study and the use of a wide range of ☐ conditions serves to better understand the physics. However, according to a recent observational based data set of CCN

(Choudhury and Tesche, 2023), in the boundary layer (height of 500 m above the surface) the 5th to 95th percentile range of ☐ (evaluated in this case at 0.2% supersaturation) is 19.9 to 1943.7 ☐ based on the global coverage and all the times available, i.e., almost exactly the range that is simulated here. Following this comment we have added the following to the revised methods : "☐ *ranges from 20 to 2000* ☐ *..., following a recent observational data set (Choudhury and Tesche, 2023), which showed the feasibility of this* ☐ *range.*"

In addition, we note that other references reported such low ☐ conditions (<20 ☐) in remote locations (Flores et al., 2021).

3. Could lower aerosol radiative sensitivities at warmer temperatures be explained simply by an overall reduction in clouds (both anvil and low cloud fractions decrease at warmer temperatures), and thus a reduction in the domain-averaged effects of aerosol-cloud interactions?

**Reply:** We thank the reviewer for this comment. Indeed, SST-driven change in the baseline cloud fraction plays a significant role in the SST-sensitivity of the response to an increase in ☐. This is now demonstrated and explained in the revised manuscript. Please see our answer to comment 1 above.

4. Intuitively, I would expect more smaller ice crystals in polluted deep convective clouds. These would, at least holding everything else constant, lead to an increase in the anvil cloud fraction. Do anvil microphysical properties change under high aerosol conditions?

   If the above is true, these changes seem to play only a second order role compared to the stability iris mechanism.

   Does this mean that the aerosol-cloud interaction community shouldn't overemphasize the direct effects of aerosols on cloud properties, but rather think about cloud adaptations?

   Moreover, the thermodynamic changes that drive these adjustments may be easier to understand (or at least more accessible to a broader community). Another implication may be that we shouldn't worry too much about getting all the microphysical details right.

**Reply:** Thank you. Indeed, increasing aerosol concentration could increase the lifetime of anvils, as was previously shown, for example by Grabowski and Morrison, 2016. In our simulations, under high aerosol conditions there are smaller ice crystals, which sediment slower from the cloud (ice sedimentation flux becoming more positive, see Fig. S11, SI below), thus acting to increase the anvil's life time. However, in this paper we show that ice cloud fraction decreases with aerosols, i.e., the opposite effect. Thus, we can conclude that in our simulations the smaller sedimentation flux plays a secondary role in ice cloud fraction changes with aerosols.

The figure below was added to the SI and a clarification was added to the manuscript: "*In addition to modifying ☐, an increase in ☐ also affects the lifetime of anvil clouds by perturbing the sedimentation rate (Grabowski and Morrison, 2016). Specifically, high aerosol conditions lead to smaller ice crystals, which sediment slower from the cloud (i.e. the sedimentation flux becomes less negative; Fig. S11, SI), thus acting to increase ☐. However, Fig. 5a shows a decrease in ☐ with ☐ in our simulations, thus making this to be only a secondary effect compared with the effect of ☐ (agreeing with previous results regarding the effect of warming on anvil clouds (Beydoun et al., 2021))*"

**Figure S11.** Domain and time mean vertical profiles of ice sedimentation flux for the cleanest run for each SST (☐ = 20 ☐; **a-e**), and its response to an increase in ☐ relative to the cleanest run for each SST (**f-j**).

Regarding the reviewer's comment about the fact that these results suggest that the aerosol-cloud interaction community shouldn't overemphasize the direct effects of aerosols on cloud properties, but rather think about cloud adaptations, we very much liked this idea and found it intriguing. However, we have decided to wait for the RCEMIP-ACI results to see if this is a robust feature of all models in order to make this claim in a paper.

5. Several plots should be improved:

   -Fig. 1 and 4: I suggest adding markers for better line visibility (or thicker lines, or both)

   -Fig. 2,3,5,8,9: add zero line where appropriate (where anomalies cross it)

   -Fig. 3,5: please use same x-axis limits if possible.

   **Reply:** Thank you. Following your suggestions, we thickened the line in Fig. 1 and 4 (now Figs. 1 and 2, respectively), added zero dashed lines in Figs. 3, 5, 8, 9 (now Figs. 3, 7, 10, 11 respectively) where appropriate, and used the same x-axis in Fig. 3. We didn't use the same x-axis in Fig. 5 (now Fig. 7) since the axes have different orders of magnitude between each other. Fig. 2 was removed due to revisions in the manuscript.

**Specific comments:**

1. Page 1, line 5:
   "decline in TOA energy gain"
   I think this can be written in simpler words

**Reply:** The sentence was rewritten for clarity: "*Notably, changes in cloud radiative effects for both the longwave and shortwave parts of the spectrum lead to a decrease in top-of-atmosphere (TOA) energy gain with increasing ☐.*"

2. Page 2, line 57:
   I suggest to remove "general"

**Reply:** The word "general" has been removed.

3. Page 3, line 80:
   Something is odd/missing there.

**Reply:** Thank you, the sentence was rewritten for clarity: "*While observations, global climate models and high-resolution, convective-permitting models predict an increase in altitude of anvil clouds while maintaining nearly fixed temperatures, they also anticipate a decrease in anvil cloud coverage with rising surface temperatures.*"

4. Page 4, line 89-90:
   Changes in ozone heating could also influence anvils (e.g. Harrop and Hartmann, 2012; Seidel and Yang, 2022)

**Reply:** Thank you. We agree with the reviewer. Following this comment a caveat was added to the methods section: "*A fixed ozone profile, representing a typical tropical atmosphere, is used here. We note that using a fixed ozone profile under different SST is not entirely realistic and may have some effect on the clouds' development (Harrop and Hartmann, 2012; Seidel and Yang, 2022).*"

5. General comment on the introduction: are there similar studies that looked at aerosol impacts in RCE? If yes, please mention them, if not, please state that this is the first one looking at it.

**Reply:** Several other studies looked at aerosol impacts in RCE, and following your comment are mentioned in the introduction: "*This is done following previous studies that uses RCE to examine different aspects of ACI (van den Heever et al., 2011; Storer and van den Heever, 2013; Beydoun and Hoose, 2019; Dagan, 2022).*"

6. Page 4, Model description:
   Anvils are ice phase clouds that strongly depend also on how freezing is parameterized in the model. Please add information also about the ice phase microphysics!

**Reply:** Thank you. Following this comment we added information about the ice phase microphysics: "*Here, ice nucleation is not directly coupled to ☐ (i.e., changes in ☐ do not change the concentration of ice nucleating particles- INP), but rather depends on the temperature and the supersaturation with respect to ice(Rasmussen et al., 2002). We note that, in realistic conditions, changes in ☐ might cause changes in INP, an effect that should be addressed in future research. In our simulations, heterogeneous nucleation dominates for temperatures higher than approximately 238 K, while ice formation is dominated by homogeneous freezing for temperatures lower than approximately 233 K (Rasmussen et al., 2002). Ice nucleation directly from vapor is not considered here.*"

7. Page 4, Experimental design:
   Is the ozone heating profile fixed? If so, I suspect anvils may already be influenced by it in the warmest experiment.

**Reply:** Thank you. Please refer to our answer to comment 4.

8. Page 6, line 141:
   *"clouds become thicker"* Do the authors mean thicker in optical sense, or simply that their vertical extent is larger?

**Reply:** We meant that the clouds' vertical extent is larger, but due to the revisions made in the paper it's no longer relevant.

9. Page 7 & 8, Fig. 2 & 3:
   With the upward shift in clouds a kg of air covers a larger volume. The increase of cloud ice/upper tropospheric cloud water may therefore not be consistent with the changes in integrated amount of ice. Does ice water path change across the investigated range of SSTs?

**Reply:** Thank you for this important question. As can be seen in the figure below, ☐ changes with SST in a non-monotonic manner. Understanding this interesting trend requires further investigation

that's out of the scope of this article, which examines the effect of ☐ on cloud development under different SSTs rather than the effect of SST. Thus, this interesting question should be looked at in future research.

[Figure]

**Figure R2.** The response of domain and time mean ice water path (☐) to an increase in SST. The values are presented relative to the coldest run (SST = 290 K) for each ☐, as indicated by the ☐ sign.

10. Page 9, lines 193-194:
    Do changes in graupel production really explain the warming in the upper troposphere? Peak warming seems to be at higher altitudes.

**Reply:** At 240-220K level, graupel and snow increase with ☐. At these temperature levels we also see an increase in latent heating (Fig. S8, SI). In higher altitudes, the air density drops. Thus, a given amount of latent heating will cause a bigger temperature change at higher altitudes than low altitudes (Gasparini et al., 2023), which can explain why peak warming seems to happen at higher altitudes than peak latent heating release. Following this comment a clarification was added: " *In addition, in higher*

*altitudes, the air density drops. Thus, a given amount of latent heating will cause a larger temperature change at higher altitudes than low altitudes (Gasparini et al., 2023)."*

11. Page 10, lines 205-208:
    It may be useful to decompose the within atmospheric heating to the cloudy and clear-sky parts. Or just clearly state that the total heating and the anomalies described here are driven by clear-sky radiative cooling.

**Reply:** Thank you. As was shown above, ☐ reduces with ☐, driving an increase of the amount of clear-sky grid points. Thus, LWC is indeed driven by clear-sky radiative cooling, which is in turn driven by changes in the cloud fraction. The additions to the manuscript following this comment: "*The enhanced* ☐ *with* ☐ *is driven by clear-sky radiative cooling, which is in turn driven by the decreased* ☐ *with* ☐ *across SSTs, as illustrated in Fig. 5a. The enhanced* ☐ *is also consistent with the reduction in* ☐ *presented in Fig. 1b.*"

12. Page 11, Fig. 9:
    How should we interpret the increase in advective tendency of the liquid/ice water static energy?

**Reply:** Thank you for this question. The increase in advective tendency of the liquid/ice water static energy is a response to an increase in latent heating. For a certain height, an increase in latent heating drives an increase in the amount of heat that will be advected from this height.

13. Page 16:
    The authors could mention two more caveats:
    a. The assumed sensitivities do not include sensitivities related to changes in ice nucleation and freezing. A polluted environment would most likely lead not only to large numbers of CCN, but also to large numbers of ice nucleating particles and/or ice nucleating particles that freeze at warmer temperatures. This could lead to significant climate impacts that could easily match those described in this paper.

**Reply:** Thank you. Please refer to our answer to comment 6.

b.  Although tropical oceanic convection doesn't have as strong a diurnal cycle as land convection, its effects cannot be neglected. If the main aerosol-mediated changes in SW radiation are indeed those in deep convective cores, they may be muted in simulations with a diurnal cycle due to the early morning peak in deep convective activity (e.g. Nesbitt and Zipser, 2003; Gasparini et al. 2022).

**Reply:** Thank you, a caveat was added to the methods section: "*A diurnal cycle is not considered here, and we note that it might affect the convective development to some extent even over the ocean (Nesbitt and Zipser, 2003; Gasparini et al., 2022).*"

The additions to the revised manuscript:

[revised manuscript text omitted]

2. In multiple locations, a distinction is made between the cloud radiative effect specifically and the overall changes to radiative effects generally. But aerosols were only allowed to change cloud microphysics and not interact with radiation directly, and thus the CRE was effectively the only thing that was allowed to change in the simulations. Indeed, the top and bottom rows in Fig. 1 appear nearly identical. Specifically, the following statements are misleading and should be reworded:

   a. L5: "we show that increasing $N_a$ leads to a decline in top-of-atmosphere (TOA) energy gain across SSTs due to changes in the cloud radiative effect"

   b. L130: "the CRE ... is identified as the main driver of ΔR variations, while changes in clear sky radiation has a minimal impact"

   c. L218: "increasing $N_a$ decreases top-of-atmosphere (TOA) energy gain ... as a result of changes in the cloud radiative effect."

   In all cases, I suggest it be made clear that the aerosol concentration affects the CRE, which is the only component of the overall aerosol radiative effect that is examined.

**Reply:** Thank you for this comment. Indeed, aerosols were allowed to affect clouds but not directly affect radiations (i.e., aerosol-radiation interactions are excluded). Following the reviewer comment, we made sure that this is clear through the manuscript.

  a. We understand the reviewer's point, but we think that mentioning that aerosol-radiation interactions are excluded doesn't belong in the abstract. We explicitly mentioned this important point in the next two instances.

  b. This sentence was changed to: "*Moreover, the CRE (calculated as all sky radiative flux minus clear sky radiative flux) is identified as the main driver of $\Delta R$ variations, while changes in clear sky radiation has a minimal impact, as indicated by Fig. 1d-f. This is true in our simulations as changes in $N_a$ do not directly affect radiation by aerosol-radiation interactions.* "

  c. This sentence was changed to: "*...we demonstrate that increasing $N_a$, which does not directly interact with radiation here, decreases top-of-atmosphere (TOA) energy gain*"

However, please note that as the thermodynamic conditions are changing with $N_a$ (Fig. 10 in the main text), the clear-sky radiation is allowed to change, and in fact changes with $N_a$ to some degree. The decomposition into cloudy and clear-sky serves to demonstrate that the clear-sky changes are minimal compared with the CRE changes.

3. The introduction presents a nice overview of research on how clouds respond to changes in SST, including the FAT hypothesis and the stability iris effect. There is also a nice overview of existing work on how aerosols affect cloud properties. However, while there is a clear motivation for investigating higher SST's, the reader may be left wondering how realistic the aerosol concentration scenarios used are, if and how global aerosol concentrations are expected to change under warming, and how the results of the study would modify the aerosol concentrations further. For example, the study finds increased aerosol concentrations may increase precipitation rates. One may hypothesize that such increased precipitation would feed back into the aerosol concentration as the aerosols get rained out faster, and thus moderate the concentration. Even if these effects are highly uncertain, they could be discussed.

**Reply:** The reviewer is correct that an increase in precipitation will feedback on the aerosol concentrations and that this feedback is not included in this study. Following this comment, a caveat was added to the revised conclusions section: "*We note that an increased surface precipitation could mean*

*that aerosols get rained out faster, thus moderating the aerosol concentration. In our simulations $N_a$ is prescribed, thus this feedback is disabled. This feedback should be examined in future studies."*

In addition, the use of a wide range of $N_a$ conditions serves to better understand the physics. To demonstrate that the range of $N_a$ conditions considered here is relevant to the real world, we use a new observational-based data set of CCN (Choudhury and Tesche, 2023), which states that in the boundary layer (at 500 m above the surface) the 5th to 95th percentile range of $N_a$ (evaluated in this case at 0.2% supersaturation) is 19.9 to 1943.7 $\mathrm{cm}^{-3}$ based on the global coverage and all the times available, i.e., almost exactly the range that is simulated here. Following this comment we have added the following to the revised methods section : *"$N_a$ ranges from 20 to 2000 $\mathrm{cm}^{-3}$ (20, 100, 200, 1000, and 2000 $\mathrm{cm}^{-3}$), following a recent observational data set (Choudhury and Tesche, 2023), which showed the feasibility of this $N_a$ range."*

4. The supplement presents plots in the same format as the plots in the main text, but every plot has all combinations of SST and $N_a$. Including all 25 profiles on each plot makes them impossible to read. I would create a new plot for each value of $N_a$ so that these plots can be read. The plots would be exactly as the top row (or only row) in Figs. 1,3,5,8,9, but for different $N_a$.

**Reply:** Thank you. Following this comment, the figures in the SI have been adjusted as suggested, but with a new plot for each SST instead for each $N_a$.

5. Most vertical profiles in the study are plotted with temperature as the vertical coordinate. I find this quite useful in some cases, e.g. fig 2a appears to support the FAT hypothesis. In other cases, it is slightly more confusing, e.g. in determining the altitude of rain production in figs. 3b and 3g. The only change I would request here is that there appears to be an artifact in most profiles that could be removed. For example, the cloud ice profile (Fig 3c) is bivalued between ~ 200k and ~ 230K. I assume one value is from the stratosphere while the other is from the troposphere. I suggest truncating the profiles where the temperature begins to increase, thus ensuring each T is uniquely associated with a single z.

**Reply:** Thank you. Following this comment we have updated all of the relevant figures in both the SI and the article, to ensure each T is uniquely associated with a single height. This was done by cutting the vertical profiles at the tropopause, defined as the level at which the temperature starts to rise with altitude.

**Specific comments**

1. L5: ", even at equilibrium conditions.": Things not at equilibrium include the SST with insolation and $N_a$ with aerosol removal via deposition. I suggest explicitly stating the equilibrium is RCE.

**Reply:** Thanks. We have followed the reviewer's suggestion and updated this sentence as follows: "*Our results indicate that ACIs are SST-dependent even under RCE conditions.*"

2. L133: Please describe how grid points are defined as cloudy for the purpose of calculating cloud fraction

**Reply:** Please see our reply to general comment 1 and the new used definition for CF.

3. L135: Does "total water" refer only to total liquid and ice? If so, please clarify. "total water" to me reads as including water vapor.

**Reply:** Thank you. Indeed, "total water" refers to liquid and ice and not to vapor, i.e., total condensed water. Due to revisions made in the paper we no longer use total water, hence this is no longer relevant.

4. L146: " under lower SST ... an increase in $N_a$ can completely suppress warm rain (see Fig. 3g)." This appears unsupported in the plot. The cleanest 290K run shows a maximum for rain of ~ 0.004g kg$^{-1}$ , while the change in rain in the polluted scenario shows a maximum change of ~ −0.002g kg$^{-1}$ . Please quantify this statement.

**Reply:** Thank you for raising this discrepancy. In the coldest simulations, a rise in $N_a$ suppressed about 30% of the rain in the domain-mean vertical profile, while in the hotter simulations a rise in $N_a$ suppressed about 10% of the rain in the entire domain. This calculation was done by integrating the rain vertical profiles of each simulation, followed by determining the difference between the most polluted ($N_a$ = 2000 cm$^{-3}$) and the cleanest ($N_a$ = 20 cm$^{-3}$) runs, relative to the cleanest run, for each SST. Following this comment we updated the sentence: "*under lower SSTs... an increase in $N_a$ can inhibit warm rain (see Fig. 3g).*"

5. L151: "The stronger increase in total water with $N_a$ under lower SSTs leads to the stronger SW reflectivity,": I'm not sure that just the result of increased total water is enough to establish causality here, with the role of anvils and changes to cloud morphologies uncertain. Perhaps just "consistent with".

**Reply:** Thank you. Due to revisions made in the paper this is no longer relevant.

6. L156: "higher $N_a$ and lower SSTs ... thus producing more graupel (Fig. 3i)" In Fig. 3i, the coldest run shows the smallest increase in graupel production. The largest increase is from the middle 300K simulation, so there is not a clear trend.

**Reply:** Thank you for this comment. Following this comment, the sentence was rewritten to better describe the results presented in the figure: "*Beside resulting in an increase in $\mathcal{L}$, the warm rain inhibition under higher $N_a$ results in more super-cooled water (Carrió et al., 2011; Chen et al., 2017, Fig. 3f), leading to higher production of snow (Chen et al., 2017, Fig. 3j), and drives higher riming rates, thus producing more graupel (Chen et al., 2017, Fig. 3i).*"

7. L169: "A reduction in $D_r$ with $N_a$ could be attributed to changes in $Q_r$": For me, it's less intuitive that $N_a$ would affect $Q_r$ than $D_r$. Throughout this section, it is not clear to me that causality can be established, though I don't understand the calculation. Please explain the calculation and consider whether it establishes causality.

**Reply:** Thank you. Throughout the section we explain that changes in $D_r$ with $N_a$ can be driven by changes in either $Q_r$ or $S$ with $N_a$, or both. To realize which ($S$ or $Q_r$) is responsible for the changes in $D_r$ with $N_a$ for the different SSTs, we calculate in Fig. 9 changes in $D_r$ with $N_a$, assuming that either $S$ or $Q_r$ are fixed at a reference $N_a$ of 200 $\text{cm}^{-3}$. That is to say that we took the vertical profile of $S$ or $Q_r$ from, for example, the simulations using SST = 300 K and $N_a$ = 200 $\text{cm}^{-3}$, and used them to calculate the vertical profiles of $D_r$ for all $N_a$ conditions under SST of 300K. Then, we examined how $D_r$ changes with $N_a$, assuming that $S$ or $Q_r$ are held fixed, for each SST. This calculation is similar to that in Bony et al., 2016, Fig. 4, but for changes in $N_a$ instead of SST. Fig. 9 shows that $D_r$ decreases monotonically with $N_a$ when $Q_r$ is held fixed, while $D_r$ changes without a clear trend with $N_a$ when $S$ (or T) is held fixed. Since $D_r$ decreases with $N_a$ for a fixed $Q_r$ but not when $S$ is held fixed, a causality between the observed changes in $S$ and the changes in $D_r$ with $N_a$ can be established.

8. . L178: "for a given SST, an increase in $N_a$ drives strong warming of the upper troposphere and a weak cooling of the lower troposphere.": The two coldest simulations do not appear to cool in the lower troposphere. They warm in the upper troposphere, but both have small regions of cooling in the upper troposphere.

**Reply:** Thank you, the sentence was rewritten to better match the data presented: "*This figure illustrates that, for a given SST, an increase in $N_a$ drives strong warming of the upper troposphere and in some cases a weak cooling of the lower troposphere.*"

9. L179: "the increase in S with $N_a$ (Fig. 5a)": Fig. 5a does not show the change with $N_a$, only the values for the cleanest simulation.

**Reply:** Thank you. The line was rewritten to better match the data presented: "*This trend demonstrates an increase in $S$ with $N_a$, which in turn explains the reduction in the anvil cloud fraction.*"

10. L190: "driven by a stronger latent heat release,": As well as weaker advection (since the advection term is negative), which appears to be a similar magnitude.

**Reply:** Thank you, the sentence was rewritten: "*...mostly driven by a stronger latent heat release…*"

11. L190: "is in turn driven by higher production rates of graupel and snow ": I would say "consistent with". The causality implied here is hard to see and may flow the other direction.

**Reply:** Thank you. We agree with the reviewer, and the sentence was rewritten as suggested: "*...which is consistent with the higher production rates of graupel and snow…*"

12. L199: Kirchoff's Law implies ΔLW C ≈ −ΔRLW because net surface LW radiation flux is zero if LW surface reflection is negligible. Is this true? The plots of ΔLW C and −ΔRLW appear almost identical.

**Reply:** This is indeed true, and is stated in the following paragraph: "*The enhanced $\Delta LWC$ with $N_a$ is driven by clear-sky radiative cooling, which is in turn driven by the decreased $CF_{ice}$ with $N_a$ across SSTs, as illustrated in Fig. 5a. The enhanced $\Delta LWC$ is also consistent with the reduction in $\Delta R^{LW}$ presented in Fig. 1b.*"

13. L219: "lower SSTs, contributed mostly by a stronger SW response.": Fig. 1 shows, especially in the colder simulations, quite similar SW and LW components to the CRE.

**Reply:** The sentence was rewritten to better match the data presented: "*We also show that this effect is stronger under lower SSTs, which is consistent with the stronger increase in liquid water path ($\mathcal{L}$) with $N_a$ under lower SSTs. The ice water path ($\mathcal{I}$) and cloud fraction (CF) responses, on the other hand, are negative and consistent across SSTs.*"

14. L228: I'm not sure this is shown clearly enough to be a conclusion. Cloud morphology, overlap, and anvil coverage may also impact the SW reflection.

**Reply:** Thank you. Following your first comment and comments made by Reviewer #1, we identified the existence of shallow clouds and used a linear decomposition in order to distinguish between cloud coverage and cloud opacity. Using this decomposition, which was also done for the different identified cloud regimes, we arrived at a more reliable conclusions: "*To better understand these trends, we decompose the response of TOA energy gain ($\Delta R$) to different cloud regimes (based on 2D histograms of $\mathcal{L}$ and $\mathcal{I}$) and to contributions from changes in the cloud opacity (the opacity term) and in cloud occurrence (the shift term) based on a linear decomposition. This decomposition illustrates that most of $\Delta R$'s negative trend and its SST-sensitivity is driven by the opacity term, which in turn is driven by the SW part of the spectrum. This trend can be explained by the Twomey effect, i.e, for a given $\mathcal{L}$ and $\mathcal{I}$ the clouds become more reflective with a rise in $N_a$. The Twomey effect is proportional to the baseline CF, thus becoming stronger under lower SST for which the baseline CF is higher. The shift term, on the other hand, illustrates a compensation between a positive response in the SW and a negative response in the LW, thus producing a small net effect. Furthermore, we decompose $\Delta R$ and its components per cloud regime, which illustrates that ice and shallow clouds are the main drivers behind the opacity and shift terms trends.*"

15. L245: This paragraph generally nicely explains the limitations of the simulation. I would perhaps also add that the finding that increased $N_a$ increases precipitation could lead to moderation of $N_a$ through faster rates of wet deposition, but that here $N_a$ was prescribed.

**Reply:** Please refer to our answer to general comment 3.

Figures: 1. Fig. 7: Please explain how this is calculated. It is not obvious to me how $Q_r$ could be held fixed in any given simulation

**Reply:** Please refer to our answer to comment 7.

**Minor comments**

1. L74: "vertical pressure velocity" → "clear sky vertical pressure velocity"

**Reply:** Thank you. The line was rewritten as suggested.

2. L81: "and observations to decrease with surface temperature " → "and observations to decrease in coverage with surface temperature "

**Reply:** Thank you. The sentence was rewritten: "*While observations, global climate models and high-resolution, convective-permitting models predict an increase in altitude of anvil clouds while maintaining nearly fixed temperatures, they also anticipate a decrease in anvil cloud coverage with rising surface temperatures*"

3. L112: "solar-insulation" → "solar insolation"

**Reply:** Thank you. The line was rewritten as suggested.

4. L115: "is fixed at pre-industrial level (280 ppm)" → "is fixed at the pre-industrial level (280 ppm)"

**Reply:** Thank you. The line was rewritten as suggested.

5. LL118: "The vertical profile of $O_3$'s represents": I suggest simply $O_3$ or just "ozone"

**Reply:** Thank you. The sentence was rewritten: "*A fixed ozone profile, representing a typical tropical atmosphere, is used here.*"

6. L141: "the isotherm height" → "freezing level"

**Reply:** Thank you. The line was rewritten as suggested.

7. L160: " reduction in cloud ice at the upper troposphere (Fig. 3f),": I think you meant 3h.

**Reply:** Thank you. The sentence was rewritten: "*In addition, cloud ice declines with $N_a$, consistently across SSTs (Fig. 3h).*"

8. L199: "calculated as the TOA's LW radiation flux" → "calculated as the TOA's net LW radiation flux". Right?

**Reply:** Thank you. The line was rewritten as suggested.

9. L227: "Hence, the liquid water content in clouds" → "Hence, the column liquid water content in clouds"

**Reply:** Thank you. Due to revisions made in the paper as mentioned in our answer to comment 14, this is no longer relevant.

**Reply:** Thank you. The line was rewritten as suggested.

**Suggestions**

The comments in this section are intended to be more suggestive. The authors are welcome to take them into consideration but need not feel obligated to.

1. L80: "Beside increasing in height": This is the first explicit mention of the prediction that anvils will rise with FAT. If a reader was not familiar with FAT, it could be nice to introduce it in the paragraph beginning at line 67.

**Reply:** The FAT hypothesis is indeed mentioned in line 66.

2. Fig. 4 and 10: Perhaps my personal preference, but I would plot on the y-axis the values rather than the difference with the cleanest run. The point would remain the same as this amounts to a vertical shift. If you choose not to, make this more clear in the captions ("relative to the cleanest run for each SST" → "Values are shown as a difference with the cleanest run for each SST." or similar)

**Reply:** Thank you. Following this suggestion the captions were rewritten for clarity: "*The values are shown as a difference between the most polluted ($N_a$ = 2000 $cm^{-3}$) and the cleanest ($N_a$ = 20 $cm^{-3}$) runs for each SST.*" We prefer to present the difference from the cleanest run in order to emphasize the effect of aerosol under different SST rather than the effect of SST.

3. I might add a thin dashed line on all plots that present changes along the $\Delta\varphi = 0$ line where $\varphi$ is the variable being plotted (similar to Fig. 10).

**Reply:** A dashed line marking the zero-value was added in the relevant figures.

4. The font sizes in plots are inconsistent. Assuming these are generated in Python, this could be fixed by setting figsize to a constant value in plt.subplots(figsize=(width, height)) and then setting the font sizes to constant values in ax.set_ylabel(), ax.set_title() etc.

**Reply:** Thank you. The font in all the figures was set to a constant value as suggested, besides in Fig. 11, where the font was made to be slightly bigger to ensure comfortable reading.

5. L182: Perhaps define $h_L$ mathematically to make clear this includes latent energy from freezing.

**Reply:** Thank you. We specified that the latent term in $h_L$ includes latent energy from freezing: "*The $h_L$ tendency equation contains 5 terms: advection (adv), radiation (rad), latent heating (includes latent heating from freezing), turbulence and large-scale tendency*"

6. L220: "pushes warm rain initiation to higher levels of the cloud" → "pushes would-be warm rain initiation to higher levels of the cloud" or something similar. In the colder simulations, these "higher levels" are above the freezing level, so rain does not form as well since the ice crystals "steal" water off the droplets that would become rain. In Fig. 3g, it is only the warmest 2 simulations that show increased rain near the freezing level. This is explained well in the following sentences.

**Reply:** Thank you. We understand the reviewer's point here but believe that this sentence is clearer in its original form since we think ".. would-be warm rain initiation…" might be hard to understand.

7. L253: "allow us to confront our conclusions": Did you mean confirm?

**Reply:** We meant that we want to check whether our conclusions are correct in a variety of models and microphysical schemes, hence the use of the word "confront" and not "confirm".

---

## Referee Report (RR1)

**Review of Lorian and Dagan, 2023, revision**

I thank the authors for significantly improving their manuscript and addressing most of my comments well. This adds much value to the manuscript. Nevertheless, I think that the new and very informative analysis is not yet at a good enough level and should therefore be revised before the manuscript can be published in final form. I also added a few more rather minor comments.

**General comments:**
1. It appears that the main radiative response is the Twomey effect. However, this effect comes mainly from ice clouds, which is plausible but rather unusual. What do such clouds look like compared to unperturbed/clean clouds? Also, to be fair, this could simply be an artifact of the lack of aerosol coupling in the freezing part of the code, which should be clearly stated somewhere.

2. The warm rain inhibition mechanism is mentioned a few times in the manuscript. Is warm rain really important under RCE conditions? Isn't most of the rain we see just from melting ice hydrometeors?

   Comments 3-8 refer to the new cloud decomposition and its implications
3. The authors follow the decomposition of Sokol et al, 2024. However, I think the way they use this decomposition is confusing, mainly because of the naming/interpretation of the individual terms.
   What the authors call the "shift term" is considered in Sokol et al., 2024 to be a combination of the area and opacity terms, plotted in black in their Fig. 3c. Sokol goes further and decomposes this into the area and opacity terms, shown in pink. It would be nice if you could do that too - but it's just a suggestion; it could be more elaborated due to your 2D phase space.

   What the authors call "opacity term" is in Sokol et al. described as
   "*The second term on the right-hand side accounts for changes in CRE(IWP), which may occur due to changes in clear-sky fluxes or cloud microphysics, temperature, and altitude. This term encompasses the entire ice cloud altitude feedback, as well as the part of the opacity feed- back related to changes in τ at fixed IWP, which may result from changes in cloud microphysical structure.*"
   So it is likely a combination of several factors. However, I would agree that intuitively most of it should come from the increased opacity at the fixed ice water path (i.e. "ice Twomey effect"). But you should mention that other factors could influence it.

   Ultimately, not much needs to change in the decomposition, but the terms need to be more clearly described, not only in comparison to Sokol et al. 2024, but also in comparison to the more widely known feedback decompositions (e.g., Zelinka et al. 2016).

4. The cloud categories limits may need to be adjusted.
   I.) In table S1, "no clouds" category goes to ice water path of 5. This limit should be corrected to at least 1 $g/m^2$, which corresponds to a cloud of a cloud optical depth of

approximately 1, which is clearly not negligible in radiative terms. The range could even go down to 0.1 g/m2, as thin clouds don't cease to exist at 1 $g/m^2$, but those thinnest clouds may be less radiatively important.

II.) Deep convective clouds occur in nature at IWP larger than about 1000 $g/m^2$. Clouds with IWP between 20 and 200 $g/m^2$ are certainly not deep convective towers (unless something is very weird/wrong in the model). Therefore, your category 4 might rather include cumulus congestus with frozen cloud tops reaching heights above 5 but below 10 km in the tropics and representing the third peak in cloud fraction (see e.g. Fig. 5a in Hartmann and Berry, or Fig. 1 in Gasparini et al., 2019).In any case, the number of the cloud regime should be added to Table S1, and the name of the cloud regime should be added to the caption of Fig. 4.

5. It would be great if the authors could add another column to your Figure 4, with values of CRE*CF, which would show the radiative significance of each ice-water path bin in your 2D space.

6. What is plotted in the first column is not cloud fraction. It is simply a 2D PDF of the frequency of occurrence (ok, cloud occurrence may be ok, but not cloud fraction). So please call it that, especially since in Figure 2 you are using the domain-averaged cloud fraction, which is something completely different.
Also, is the sum of the frequency of occurrence over the whole phase space equal to 1?

7. I am confused why the "shallow" category seems to be as radiatively important. In figure S6 we see that the shallow cloud fraction is about 0.2%, compared to the ice cloud fraction of 40% (let's assume that splits equally - 20/20 to thin and thick ice). That's 2 orders of magnitude difference. The difference in CRE (column 2 in Figure 4), however, seems to be at most 1 order of magnitude.
Why is therefore the decomposition for shallow leading to same magnitude size of effect in Figure 6? Am I missing something?

8. It's very hard to see the occurrence frequency values in column 1. Maybe plotting as pcolor instead of contourf could help? Also, is it really important to go to values as low as 0.0001? Couldn't the colormap stop at log(cf)=-3? And start maybe at -1?

**Specific comments:**
You may want to update the Sokol et al., 2024 citation; it should appear in final form in the coming days in Nat. Geosci.

Page 1, lines 6-7: What does the sentence "The changes in..." really mean?
I thought you explain the key radiative difference with the Twomey effect, not changes in cloud fraction?

Page 1, line 13:
"decline in TOA longwave energy gain"
I guess this is a very complicated way to say "more outgoing longwave radiation".

Line 61: generally => often (they are indeed not always opaque in infrared; most frequent high clouds at COD<1 are not)

Page 3, line 81: Delete Lindzen et al., 2001 and Mauritsen and Stevens, 2015 reference if you strictly describe the stability iris hypothesis. Lindzen's Iris hypothesis is different; it is a microphysical iris, and not the stability iris you describe; a similar iris formulation was also considered by Mauritsen and Stevens, 2015.

Page 4, section 2.1:
I imagine that a reference to the microphysical scheme would make more sense than the cited paper, which seems to be about processes and not parameterization. The two-moment bulk microphysics of Morrison et al. (2005) probably uses the Cooper et al., 1986 formulation for deposition freezing. Indeed, the best way to confirm this is to search the microphysics scheme in the code.

Page 4, Lines 108-110:
*In our simulations, heterogeneous nucleation dominates for temperatures higher than approximately 238 K, while ice formation is dominated by homogeneous freezing for temperatures lower than approximately 233 K (Rasmussen et al., 2002).*

I don't think that's necessarily true (unless you've checked it yourself). Homogeneous freezing of cloud droplets is only active at the homogeneous freezing temperature of water, not below/above it. So I suggest deleting this sentence and just mentioning which parameterizations are used for freezing. I assume:

1. Coper et al., 1986 for deposition freezing, which is also active at T<233 K (should not be the case in reality, but that's what the model likely does)
2. Homogeneous freezing of water droplets (no need for a reference, as it's simply a statement of the kind: "if cloud droplets present at T<233 K, freeze them")

I imagine there is no physical process that would be able to nucleate ice at T<233 K. Instead, and contrary to what is known about ice nucleation, Cooper et al., 1986 are allowed to be active at such conditions (probably along with some strong artificial limits on nucleated ice crystals to prevent the model from getting crazy numbers of ice crystals).

Page 7, line 174-176:
*"We note that the average CRE of thin anvil cloud is small but not positive as in previous assessments (Sokol, 2024), probably due to the use of a relatively coarse resolution of $\mathcal{L}$ and $\mathcal{I}$ bins"*
And what if it is because low clouds that occur below ice clouds are affecting the result?

Page 7, line 175:
Sokol et al., 2024 just analyzes RCE simulations. Other studies look at satellite retrieved CRE, and may deserve to be mentioned. E.g. Hong et al., 2015, Hong et al., 2016, Fig 1 in Gasparini et al., 2019, etc.

Page 12, lines 231-232:
The net effect of the shift term is not negligible for this ice clouds, in Fig 6.

Page 12, lines 235-248:
I thought the definition of shallow clouds is that they don't reach the freezing level. But around line 245 I see explanations that involve changes in the freezing level. Please clarify!

Page 12, title 3.3:
Please use words.

Page 16, section 3.4:
I think the paper is already dense enough that you could remove this section to keep focus on the radiative fluxes.

**References:**
Cooper et al., 1986; https://doi.org/10.1175/0065-9401-21.43.29
Gasparini et al., 2019; https://doi.org/10.1029/2019MS001736
Gasparini et al., 2022; https://doi.org/10.1175/JCLI-D-21-0211.1
Hartmann and Berry, 2017; https://doi.org/10.1002/2017JD026460
Hong et al., 2015; https://doi.org/10.1175/JCLI-D-14-00666.1
Hong et al., 2016; https://doi.org/10.1175/JCLI-D-15-0799.1
Zelinka et al., 2016; https://doi.org/10.1002/2016GL069917

Best regards,
Blaž Gasparini

---

## Author Response (AR3)

**Reply to the referee's review of: On the sensitivity of aerosol-cloud interactions to changes in sea surface temperature in radiative-convective equilibrium**

Suf Lorian[1] and Guy Dagan[1]

[1]Fredy and Nadine Herrmann Institute of Earth Sciences, The Hebrew University of Jerusalem, Jerusalem, Israel

We would like to thank the Editor and Reviewer #1 for their constructive suggestions. Please find below a point-by-point reply to all of the comments (in blue).

**1   Editor**

I would also like to see some consideration given to the recent opinion piece by Varble et al. (2023) "A critical evaluation of the evidence for aerosol invigoration of deep convection" where they cast a critical eye on any possible role of convective invigoration through latent heat release. Presumably SAM accounts for the effect on buoyancy of increased condensate loading as a counteracting force, but I don't see this mentioned.

**Reply:** Thank you. Convective invigoration refers to the case in which an increase in aerosol loading drives an increase in vertical velocity in convective clouds. In our article, we don't relate the latent heat release to changes in vertical velocity, in fact we are not dealing with vertical velocities at all, thus, making convective invigoration process irrelevant to our paper. We mention convective invigoration in the introduction section, where we specify that convective invigoration through latent heat release is highly questionable: "*Under this hypothesis, which remains highly questionable (Varble et al., 2023; Romps et al., 2023), increasing aerosol concentrations have been suggested to drive stronger latent heat release and hence stronger vertical velocities. In addition, under high aerosol concentration conditions, the smaller hydrometeors are transported higher into the atmosphere for a given vertical velocity (Koren et al., 2015; Dagan et al., 2018, 2020), and their lifetime at the upper troposphere is longer, due to a weaker sedimentation rate (Fan et al., 2013; Grabowski and Morrison, 2016). However, it is important to note that these proposed aerosol effects are still highly uncertain (Stevens and Feingold, 2009; Varble, 2018; Romps et al., 2023; Varble et al., 2023).*"

In our case we use the enhanced latent heating at the upper troposphere under polluted conditions to explain the increase static-stability, which affects the radiatively-driven divergence and thus the anvil cloud fraction. An enhanced water loading, which is accounted for in SAM and will impact the buoyancy (and hence the vertical velocity), will not directly affect the static-stability, and hence is not relevant for the mechanism explained in our paper (which is again different from "convective invigoration" as we are not examining the effects of aerosols on vertical velocities).

Also, in the conclusions, a passing mention is given to how only a small domain is considered, but not speculation given to what the intuitively might be expected were a larger domain simulated. Presumably something can be said building on prior SST sensitivity studies using a channel domain. Alternatively, there's physical or observational arguments (e.g. DeWitt et al. 2023 in ACP) that suggest aerosol loading doesn't affect distributions of cloud sizes, and hence cloud behaviors (tentatively) may be quite robust if sufficiently large time and space scales are considered.

**Reply:** Thank you. A recent study (Dagan, 2024), focusing on a larger domain simulations (long-channel domain), showed that an aerosol perturbation in these simulations intensifies the large-scale circulation, dominating the domain-mean cloud and radiative response. These effect are not accounted for in small-domain simulations as used in our current study, which focus on the local response. Following this comment, the following was added to the conclusions section: "*In a larger domain, the circulation is suggested to intensify with an increase in $N_a$ (Dagan, 2024; Dagan et al., 2023). In this case, the large-scale circulation changes dominate the change in the domain-mean cloud and radiative properties. In our simulations we are focusing on the local response and these larger-scale effects are not accounted for.*"

**2 Reviewer #1**

I thank the authors for significantly improving their manuscript and addressing most of my comments well. This adds much value to the manuscript. Nevertheless, I think that the new and very informative analysis is not yet at a good enough level and should therefore be revised before the manuscript can be published in final form. I also added a few more rather minor comments.

**Reply:** We would like to express our gratitude to the reviewer for his extremely constructive comments and the time he spent suggesting ways to improve our manuscript. It is truly remarkable how much these comments (in the previous round and in this round) have helped us improve our paper. Thank you very much!

**2.1 General comments**

1. It appears that the main radiative response is the Twomey effect. However, this effect comes mainly from ice clouds, which is plausible but rather unusual. What do such clouds look like compared to unperturbed/clean clouds? Also, to be fair, this could simply be an artifact of the lack of aerosol coupling in the freezing part of the code, which should be clearly stated somewhere.

**Reply:** Thank you. One can get a feeling of how these clouds look like from Fig. 3 from the main text (and presented below) and from the newly added snapshots of outgoing longwave radiation (OLR) to the supporting information (Fig. S1, SI below).

A reference to the snapshots was added to the methods section: "*The SST ranges from 290 to 310 K in 5 K intervals. Snapshots of the different simulations is presented in Fig. S1, SI.*"

[Figure]

**Figure S1.** Snapshots of the outgoing longwave radiation (OLR) of the different simulations.

Furthermore, a caveat was added to the results section: *This term represents an increase in the reflectivity of the ice clouds for a given $\mathcal{L}$ and $\mathcal{I}$ distribution, and can be explained by a similar mechanism to the Twomey effect but for ice particles. We note that this result might differ under coupling of $N_a$ to ice nucleating particles, which is not considered here.*

2. The warm rain inhibition mechanism is mentioned a few times in the manuscript. Is warm rain really important under RCE conditions? Isn't most of the rain we see just from melting ice hydrometeors?

**Reply:** Thank you. In our simulations, under very clear conditions (and especially under high SSTs for which the warm section is deeper) the warm rain production is not negligible. For example, under SST of 310K and $N_a$ of 20 cm$^{-3}$, almost 20% of the rain is from shallow clouds (according to the cloud regimes definition presented in the paper). Thus, an increase in aerosol loading has a potential to significantly reduce this non-negligible fraction. Indeed, as can be seen from Fig. 3 from the main text (presented below), rain is reduced in the warm phase with an increase in aerosol concentration for all SSTs. The warm section acts as the boundary and initial condition for the mixed-phase section, thus any changes in the warm phase affect the mixed phase. Specifically, the delay in warm rain formation leads to higher production rates of graupel and snow ((Chen et al., 2017); as can also be seen in Fig. 3 below).

Comments 3-8 refer to the new cloud decomposition and its implications

[Figure]

**Figure 3.** Domain and time mean vertical profiles of the different hydrometeors for the cleanest runs ($N_a = 20 \ \mathrm{cm}^{-3}$): (**a**) cloud liquid water, (**b**) rain, (**c**) ice, (**d**) graupel, and (**e**) snow, and their response to increasing $N_a$ to 2000 $\mathrm{cm}^{-3}$, relative to the cleanest run for each SST (**f** – **j**). Here we only present the cleanest and the response of the most polluted runs for clarity. The full range of $N_a$ is presented in Figs. S3-S7, SI.

3. The authors follow the decomposition of Sokol et al, 2024. However, I think the way they use this decomposition is confusing, mainly because of the naming/interpretation of the individual terms. What the authors call the "shift term" is considered in Sokol et al., 2024 to be a combination of the area and opacity terms, plotted in black in their Fig. 3c. Sokol goes further and decomposes this into the area and opacity terms, shown in pink. It would be nice if you could do that too - but it's just a suggestion; it could be more elaborated due to your 2D phase space.

What the authors call "opacity term" is in Sokol et al. described as "The second term on the right-hand side accounts for changes in CRE(IWP), which may occur due to changes in clear-sky fluxes or cloud microphysics, temperature, and altitude. This term encompasses the entire ice cloud altitude feedback, as well as the part of the opacity feed- back

related to changes in $\tau$ at fixed IWP, which may result from changes in cloud microphysical structure." So it is likely a combination of several factors. However, I would agree that intuitively most of it should come from the increased opacity at the fixed ice water path (i.e. "ice Twomey effect"). But you should mention that other factors could influence it.

80  Ultimately, not much needs to change in the decomposition, but the terms need to be more clearly described, not only in comparison to Sokol et al. 2024, but also in comparison to the more widely known feedback decompositions (e.g., Zelinka et al. 2016).

**Reply:** Thank you for this comment. Following this comment, a better explanation for each term was added: "... *we decompose the mean $\Delta R$ into three contributions:*

$$\Delta CRE \cong \Delta R$$

$$= \underbrace{\int_0^\infty \int_0^\infty \Delta CRE(\mathcal{L},\mathcal{I})CF(\mathcal{L},\mathcal{I})\,d\mathcal{L}\,d\mathcal{I}}_{Opacity}$$

$$+ \underbrace{\int_0^\infty \int_0^\infty CRE(\mathcal{L},\mathcal{I})\Delta CF(\mathcal{L},\mathcal{I})\,d\mathcal{L}\,d\mathcal{I}}_{Shift}$$

$$+ \underbrace{\int_0^\infty \int_0^\infty \Delta CRE(\mathcal{L},\mathcal{I})\Delta CF(\mathcal{L},\mathcal{I})\,d\mathcal{L}\,d\mathcal{I}}_{Nonlin} \tag{1}$$

*In this decomposition, the first term on the right-hand-side, the "Opacity" term, represents changes in $\Delta R$ due to changes in the CRE per $\mathcal{L}$ and $\mathcal{I}$ bin, while the distribution of $\mathcal{L}/\mathcal{I}$ are held fixed, i.e., this term is calculated by multiplying Fig. 4a with Fig. 4k. This term represents changes in the cloud's opacity (reflectance and absorption) for a given liquid and ice amount (for example by the Twomey effect). We note that this term could also be influenced by changes in clear-sky fluxes (Sokol et al., 2024). The second term on the right-hand-side, the "Shift" term, represents changes in $\Delta R$ due to changes in the distribution of $\mathcal{L}/\mathcal{I}$ occurrence, while the CRE per $\mathcal{L}$ and $\mathcal{I}$ bin is held fixed, i.e., this term is calculated by multiplying Fig. 4b with Fig. 4l. The Shift term is contributed by both changes in the total CF and by a shift between the different cloud regimes (for example thinning of ice clouds). The last term on the right-hand-side, the nonlinear ("Nonlin") term, represents the combined effect of changes in the CRE and the cloud occurrence in the different $\mathcal{L}/\mathcal{I}$ bins, i.e., this term is calculated by multiplying Fig. 4k with Fig. 4l.*

4. The cloud categories limits may need to be adjusted.

    (a) In table S1, "no clouds" category goes to ice water path of 5. This limit should be corrected to at least 1 g m$^{-2}$, which corresponds to a cloud of a cloud optical depth of approximately 1, which is clearly not negligible in radiative

terms. The range could even go down to 0.1 g m$^{-2}$, as thin clouds don't cease to exist at 1 g m$^{-2}$, but those thinnest clouds may be less radiatively important.

**Reply:** Thank you for this important comment. To further understand the effect of the selected boundaries of the "No Cloud" regime on the cloud fraction (CF) trend under the different simulations, we have calculated it with varying boundary definition in the range of $\mathcal{L} < (1, 2, 4, 8)$ and $\mathcal{I} < (0.5, 1, 2, 4)$ (Fig. S2, SI, presented below). As can be seen from this analysis, the trend in total CF with $N_a$ doesn't vary much for the different values of $\mathcal{L}$ described above, but exhibits different trends for the different $\mathcal{I}$ values described above. This figure is consistent with the thinning of ice clouds with an increase in $N_a$, as explained in the manuscript. Following this comment (and the following comments), the different cloud regimes' boundaries were changed and specifically, we now follow the reviewer's suggestion regarding the "No Cloud" regime. The new boundaries of the different regimes can be seen in Table S1, SI, presented below.

**Table S1.** Cloud regime's liquid water path ($\mathcal{L}$) and ice water path ($\mathcal{I}$) boundaries.

| Cloud regime | $\mathcal{L}$ [g m$^{-2}$] | $\mathcal{I}$ [g m$^{-2}$] |
|---|---|---|
| No clouds | $0<\mathcal{L}<1$ | $0<\mathcal{I}<1$ |
| 1) Thick ice | $0<\mathcal{L}<1$ | $16<\mathcal{I}$ |
| 2) Thin ice | $0<\mathcal{L}<1$ | $1<\mathcal{I}<16$ |
| 3) Shallow | $1<\mathcal{L}$ | $0<\mathcal{I}<16$ |
| 4) Deep | $1<\mathcal{L}$ | $16<\mathcal{I}$ |

(b) Deep convective clouds occur in nature at IWP larger than about 1000 g m$^{-2}$. Clouds with IWP between 20 and 200 g m$^{-2}$ are certainly not deep convective towers (unless something is very weird/wrong in the model). Therefore, your category 4 might rather include cumulus congestus with frozen cloud tops reaching heights above 5 but below 10 km in the tropics and representing the third peak in cloud fraction (see e.g. Fig. 5a in Hartmann and Berry, or Fig. 1 in Gasparini et al., 2019). In any case, the number of the cloud regime should be added to Table S1, and the name of the cloud regime should be added to the caption of Fig. 4.

**Reply:** Thank you. We completely agree with the reviewer. Following this comment a caveat was added to the manuscript: "*We note that the shallow and deep cloud regimes may also consist of other types of clouds, such as cumulus congestus in the deep regime and two-layer cloud conditions with cirrus clouds with relatively low $\mathcal{I}$ above shallow clouds.*"

Furthermore, the number of cloud regimes was added to Table S1, SI above, and the following addition was added to the caption of Fig. 4 in the manuscript: "*Four different cloud regimes are marked in red in panel **a**: (1) thick anvil clouds, (2) thin anvil clouds, (3) shallow clouds and (4) deep convective clouds, while the clear-sky regime is painted in tan.*"

[Figure]

**Figure S2.** The response of domain and time mean cloud fraction (CF) to an increase in $N_a$. The values are presented relative to the cleanest run ($N_a = 20$ cm$^{-3}$) for each SST, as indicated by the $\Delta$ sign. Four different limits of liquid water path ($\mathcal{L}$) and ice water path ($\mathcal{I}$) are considered for the "No clouds" regime to examine its sensitivity.

5. It would be great if the authors could add another column to your Figure 4, with values of CRE*CF, which would show the radiative significance of each ice-water path bin in your 2D space.

**Reply:** Thank you for this suggestion. Following this comment and comments 6 and 8, Fig. 4 was revised as suggested in these comments (see below).

6. What is plotted in the first column is not cloud fraction. It is simply a 2D PDF of the frequency of occurrence (ok, cloud occurrence may be ok, but not cloud fraction). So please call it that, especially since in Figure 2 you are using the domain-

[Figure]

**Figure 4.** Domain and time mean two-dimensional histograms of cloud occurrence (CO; **a** and **f**), at different bins of liquid water path ($\mathcal{L}$) and ice water path ($\mathcal{I}$) and the average total (**b** and **g**), shortwave (**c** and **h**) and longwave (**d** and **i**) cloud radiative effect (CRE) at these different bins. Furthermore, the radiative significance (CRE×CO) of each bin is illustrated in panels **e** and **j**. These quantities are presented for two simulations using the lowest ($N_a = 20$ cm$^{-3}$; **a-e**), and the highest ($N_a = 2000$ cm$^{-3}$; **f-j**) $N_a$, under SST = 290 K. Four different cloud regimes are marked in red in panel **a**: (1) thick anvil clouds, (2) thin anvil clouds, (3) shallow clouds and (4) deep convective clouds, while the clear-sky regime is painted in tan. In addition, the difference between the highest and lowest $N_a$ conditions is presented in panels **k-n**.

130   averaged cloud fraction, which is something completely different. Also, is the sum of the frequency of occurrence over the whole phase space equal to 1?

**Reply:** Yes, the sum of the occurrence over the whole phase space equals to 1. All instances of cloud fraction (CF) when referring to the histograms were changed to cloud occurrence (CO) as the reviewer suggested.

7. I am confused why the "shallow" category seems to be as radiatively important. In figure S6 we see that the shallow cloud fraction is about 0.2%, compared to the ice cloud fraction of 40% (let's assume that splits equally - 20/20 to thin and thick ice). That's 2 orders of magnitude difference. The difference in CRE (column 2 in Figure 4), however, seems to be at most 1 order of magnitude. Why is therefore the decomposition for shallow leading to same magnitude size of effect in Figure 6? Am I missing something?

   **Reply:** Thank you for spotting it. Following this comment, a mistake in the code calculating shallow CF was found, and is now fixed. Fig. S8, SI (presented below) illustrates that shallow clouds consist of 4.5-6.5% of the domain, i.e., an order of magnitude smaller than ice clouds. Even with these differences in CF, shallow clouds have a more negative CRE (Fig. 4 b and g), and are susceptible to changes in $N_a$ (Fig. 4l), thus contributing a significant amount to the changes in $\Delta R$.

8. It's very hard to see the occurrence frequency values in column 1. Maybe plotting as pcolor instead of contourf could help? Also, is it really important to go to values as low as 0.0001? Couldn't the colormap stop at log(cf)=-3? And start maybe at -1?

   **Reply:** Thank you. We have followed these suggestions. Please refer to our answer to general comment 5.

**2.2 Specific comments**

1. You may want to update the Sokol et al., 2024 citation; it should appear in final form in the coming days in Nat. Geosci.

   **Reply:** Thank you, the citation was updated as suggested.

2. Page 1, lines 6-7: What does the sentence "The changes in..." really mean? I thought you explain the key radiative difference with the Twomey effect, not changes in cloud fraction?

   **Reply:** Thank you. Indeed, we explain the key radiative effect with the Twomey effect, which is dependant on the baseline cloud fraction. Specifically, the larger the baseline cloud fraction, the larger the magnitude of the Twomey effect. The baseline CF decreases with an increase in SST (Fig. S8, SI). Following this comment, the abstract was revised: "*The changes in TOA shortwave flux exhibit greater sensitivity to underlying SST conditions compared to longwave radiation. To comprehend these trends, we perform a linear decomposition, analyzing the responses of different cloud regimes and contributions from changes in cloud's opacity and occurrence. This breakdown reveals that ice and shallow clouds predominantly contribute to the radiative effect, mostly due to changes in cloud's opacity, due to the Twomey effect, which is proportional to the baseline cloud fraction.*"

3. Page 1, line 13: "decline in TOA longwave energy gain" I guess this is a very complicated way to say "more outgoing longwave radiation".

   **Reply:** Thank you, the sentence was reworded as suggested: "... *iris–stability effect, resulting in an increase in outgoing longwave radiation.*"

[Figure]

**Figure S8.** Changes in domain and time mean cloud fraction of thick ice ($CF_{thick}$; **a**), thin ice ($CF_{thin}$; **b**), shallow ($CF_{shallow}$; **c**) and deep convective clouds ($CF_{deep}$; **d**) due to an increase in $N_a$, for each SST.

4. Line 61: generally => often (they are indeed not always opaque in infrared; most frequent high clouds at COD<1 are not)

   **Reply:** Thank you, "*generally*" was changed to "*often*" as suggested.

5. Page 3, line 81: Delete Lindzen et al., 2001 and Mauritsen and Stevens, 2015 reference if you strictly describe the stability iris hypothesis. Lindzen's Iris hypothesis is different; it is a microphysical iris, and not the stability iris you describe; a similar iris formulation was also considered by Mauritsen and Stevens, 2015.

 **Reply:** Thank you, the articles mentioned above were removed following this comment.

6. Page 4, section 2.1: I imagine that a reference to the microphysical scheme would make more sense than the cited paper, which seems to be about processes and not parameterization. The two-moment bulk microphysics of Morrison et al. (2005) probably uses the Cooper et al., 1986 formulation for deposition freezing. Indeed, the best way to confirm this is to search the microphysics scheme in the code.

**Reply:** Thank you. The microphysics scheme used in the model is indeed the one described in Morrison et al. (2005). The cited paper of Morrison et al. (2005) is indeed about the parameterization and its description (and not about processes, as the reviewer suggested).

7. Page 4, Lines 108-110: In our simulations, heterogeneous nucleation dominates for temperatures higher than approximately 238 K, while ice formation is dominated by homogeneous freezing for temperatures lower than approximately 233 K (Rasmussen et al., 2002).

I don't think that's necessarily true (unless you've checked it yourself). Homogeneous freezing of cloud droplets is only active at the homogeneous freezing temperature of water, not below/above it. So I suggest deleting this sentence and just mentioning which parameterizations are used for freezing. I assume:

   (a) Cooper et al., 1986 for deposition freezing, which is also active at T<233 K (should not be the case in reality, but that's what the model likely does)

   (b) Homogeneous freezing of water droplets (no need for a reference, as it's simply a statement of the kind: "if cloud droplets present at T<233 K, freeze them")

   I imagine there is no physical process that would be able to nucleate ice at T<233 K. Instead, and contrary to what is known about ice nucleation, Cooper et al., 1986 are allowed to be active at such conditions (probably along with some strong artificial limits on nucleated ice crystals to prevent the model from getting crazy numbers of ice crystals).

   **Reply:** Thank you. The cited paper should've been Morrison et al. (2005) and not Rasmussen et al. (2002). Following this comment, this sentence was revised: "*In our simulations, freezing occurs through homogeneous freezing and heterogeneous freezing by contact or immersion freezing (Morrison et al., 2005). Ice nucleation directly from vapor is not considered here, but depositional growth of cloud ice is enabled.*"

8. Page 7, line 174-176: "We note that the average CRE of thin anvil cloud is small but not positive as in previous assessments (Sokol, 2024), probably due to the use of a relatively coarse resolution of and bins" And what if it is because low clouds that occur below ice clouds are affecting the result?

**Reply:** Thank you. Following this comment and some previous ones, we have conducted these calculations with higher resolution of $\mathcal{L}$ and $\mathcal{I}$ bins. The new calculation with the higher resolution shows positive CRE of thin anvil clouds, as known from previous studies. Thus, this sentence was removed.

9. Page 7, line 175: Sokol et al., 2024 just analyzes RCE simulations. Other studies look at satellite retrieved CRE, and may deserve to be mentioned. E.g. Hong et al., 2015, Hong et al., 2016, Fig 1 in Gasparini et al., 2019, etc.

   **Reply:** Following the previous comment, this sentence was removed.

10. Page 12, lines 231-232: The net effect of the shift term is not negligible for this ice clouds, in Fig 6.

   **Reply:** Thank you. Following this comment and changes to the bin's resolution, the shift term of ice cloud is indeed not negligible, and a new explanation was added: "*Thin ice clouds exhibit an opposite trend to thick clouds in the shift term, due to them increasing in CO with an increase in $N_a$ (Figs. 4k and 5b). However, the combined net effect of thick and thin ice clouds on the shift term is low, due to them being similar in magnitude but opposite in sign (Fig. 6d).*"

11. Page 12, lines 235-248: I thought the definition of shallow clouds is that they don't reach the freezing level. But around line 245 I see explanations that involve changes in the freezing level. Please clarify!

   **Reply:** Indeed the definition of shallow clouds in our case is that they have little ice water path and thus are likely to not reach the freezing level. However, the freezing level becomes deeper with an increase in SST. Thus, under higher SSTs shallow cloud (which do not reach the freezing level) can form deeper. In these relatively deeper, but still warm, clouds, forming under high SST, warm rain inhibition by aerosols is less likely (the clouds are deep enough so that warm rain in inhibited at the lower part of the clouds but is formed at higher, still warm, sections of the clouds). To make this clearer, we have replaced "freezing level" with "warm layer depth" in this section:

   "*The contrasting response of $CF_{shallow}$ to $N_a$ under the different SSTs can be explained by warm rain inhibition at varying depths of warm layers. As was noted above, with an increase in SST, the warm layer depth increases, while an increase in $N_a$ acts to push warm rain formation to higher levels (Rosenfeld, 2000; Freud and Rosenfeld, 2012; Heikenfeld et al., 2019). Thus, under lower SSTs, for which the warm layer depth is relatively shallow, an increase in $N_a$ can inhibit warm rain (see Fig. 3g) and hence lead to an increase in $CF_{shallow}$. In contrast, under higher SSTs, for which the warm layer depth is relatively deep, an increase in $N_a$ drives warm rain inhibition at the lower levels, which is compensated for at higher levels of the warm section (Fig 3g), thus eliminating the positive effect on $CF_{shallow}$.*"

12. Page 12, title 3.3: Please use words.

   **Reply:** The title was changed to: "Mechanism behind the ice cloud fraction's response to $N_a$".

13. Page 16, section 3.4: I think the paper is already dense enough that you could remove this section to keep focus on the radiative fluxes.

   **Reply:** Thank you. Although the paper might seem dense, we feel that this section adds important physical understanding. Considering the section's short length, we've decided to keep it in the manuscript.

**References**

Chen, Q., Koren, I., Altaratz, O., Heiblum, R. H., Dagan, G., and Pinto, L.: How do changes in warm-phase microphysics affect deep convective clouds?, Atmospheric Chemistry and Physics, 17, 9585–9598, https://doi.org/10.5194/acp-17-9585-2017, 2017.

Dagan, G.: Large-Scale Tropical Circulation Intensification by Aerosol Effects on Clouds, Geophysical Research Letters, 51, e2024GL109 015, https://doi.org/10.1029/2024GL109015, e2024GL109015 2024GL109015, 2024.

Dagan, G., Yeheskel, N., and Williams, A. I.: Radiative forcing from aerosol–cloud interactions enhanced by large-scale circulation adjustments, Nature Geoscience, https://doi.org/10.1038/s41561-023-01319-8, 2023.

Freud, E. and Rosenfeld, D.: Linear relation between convective cloud drop number concentration and depth for rain initiation, Journal of Geophysical Research: Atmospheres, 117, 2012.

Heikenfeld, M., White, B., Labbouz, L., and Stier, P.: Aerosol effects on deep convection: the propagation of aerosol perturbations through convective cloud microphysics, Atmospheric Chemistry and Physics, 19, 2601–2627, 2019.

Morrison, H., Curry, J., and Khvorostyanov, V.: A new double-moment microphysics parameterization for application in cloud and climate models. Part I: Description, Journal of the atmospheric sciences, 62, 1665–1677, 2005.

Rasmussen, R. M., Geresdi, I., Thompson, G., Manning, K., and Karplus, E.: Freezing Drizzle Formation in Stably Stratified Layer Clouds: The Role of Radiative Cooling of Cloud Droplets, Cloud Condensation Nuclei, and Ice Initiation, Journal of the Atmospheric Sciences, 59, 837 – 860, https://doi.org//10.1175/1520-0469(2002)059<0837:FDFISS>2.0.CO;2, 2002.

Rosenfeld, D.: Suppression of rain and snow by urban and industrial air pollution, science, 287, 1793–1796, 2000.

Sokol, A. B., Wall, C. J., and Hartmann, D. L.: Greater climate sensitivity implied by anvil cloud thinning, Nature Geoscience, https://doi.org/10.1038/s41561-024-01420-6, 2024.